# Iron's role in soil organic carbon (de)stabilization in mangroves under land use change

Francisco Ruiz [1] ✉, Angelo Fraga Bernardino [2], Hermano Melo Queiroz[3], Xosé Luis Otero [4,5], Cornelia Rumpel [6] & Tiago Osório Ferreira [1,7,8] ✉

Mangroves are coastal hotspots for carbon storage and yet face multiple threats from anthropogenic activities. Here we explore the role of iron-mediated organomineral interactions (FeOMIs) in soil organic carbon (SOC) stabilization and their sensitivity to land use change (LUC) in Amazonian mangroves. We show that Fe oxides protect more labile SOC fractions, which would otherwise be vulnerable to biological degradation, with poorly crystalline Fe oxides being the most effective phase for SOC retention. Despite the fragile equilibrium of FeOMI under dynamic redox conditions in mangroves, this balance sustains approximately 8% of total SOC. The studied LUC scenario led to massive loss of FeOMIs as less crystalline phases were either degraded or transformed into more crystalline ones, losing the efficiency in retaining SOC. The conversion of mangroves to pastures and shrimp ponds, which are pervasive globally, triggers important biogeochemical changes, with major implications for the carbon sequestration potential of mangrove soils.

Mangrove forests are unique ecosystems offering a variety of key ecosystem services, including nutrient cycling, provision of habitat, food for numerous species, coastal protection, and carbon storage[1]. These forests store considerable amounts of carbon−11.7 petagrams (Pg), with 10.2 Pg (87%) stored in their soils[2]. Despite their carbon storage capacity, which occurs under harsh environmental conditions (e.g., high salinity, anoxia, tidal flooding)[3], mangroves face substantial threats from human activities[4]. Land use change (LUC) alone has been responsible for the loss of more than 180,000 hectares of mangrove forests in the past two decades[5]. Accordingly, the conversion of mangrove forests to agricultural, aquacultural, and urban areas led to the loss of 158 million tons of carbon stocks[6].

The Amazonian mangroves are recognized as some of the most carbon-rich forests in the world as their ability to accumulate and retain carbon surpasses up to four times that of the upland Amazon Forest[7]. However, the Amazonian mangroves face an alarming loss rate of 0.12% per year, representing the emission of nearly 1 Tg $CO_2$e $yr^{-1}$ [8]. The loss of soil organic carbon (SOC) stocks is of special concern since SOC is considered the most stable pool of ecosystem carbon that may persist for centuries to millennia[9,10]. While the impact of LUC on C stocks of mangroves is well-documented[2,11], few studies have explored the biogeochemical processes involved in the (de)stabilization of organic carbon in mangrove soils[12]. As a result, our understanding of the key mechanisms driving C stock losses in these forests remains incomplete, hindering the establishment of effective conservation,

[1]Department of Soil Science, "Luiz de Queiroz" College of Agriculture/University of São Paulo (ESALQ/USP), Av. Pádua Dias 11, Piracicaba, São Paulo, Brazil. [2]Department of Oceanography, Federal University of Espírito Santo, Vitória, Espírito Santo, Brazil. [3]Department of Geography, University of São Paulo, Cidade Universitária, Av. Prof. Lineu Prestes, 338, São Paulo, SP, Brazil. [4]CRETUS, Department of Edaphology and Agricultural Chemistry, Faculty of Biology, University of Santiago de Compostela – USC, Rúa Lope Gómez de Marzoa, s/n. Campus sur, Santiago de Compostela, Spain. [5]REBUSC Network of Biological Stations of the University of Santiago de Compostela, A Graña Marine Biology Station, Ferrol, Spain. [6]CNRS, Institute for Ecology and Ecological Sciences, (iEES Paris, UMR Sorbonne Université, CNRS, INRAE, UPEC, IRD), Paris, France. [7]Center for Carbon Research in Tropical Agriculture (CCARBON) - University of São Paulo, Piracicaba, São Paulo, Brazil. [8]Research Centre for Greenhouse Gas Innovation (RCGI), University of São Paulo, Av. Professor Mello Moraes 2231, São Paulo, SP, Brazil. ✉e-mail: francisco.ruiz@usp.br; toferreira@usp.br

restoration, and adaptation strategies to mitigate loss and enhance recovery of SOC.

SOC stabilization in mangroves, as in other ecosystems, is likely influenced by the complex interplay between organic substrates, microorganisms, environmental variables (e.g., temperature, moisture), and soil physico-chemical properties[13]. Traditionally, factors such as high primary production, high recalcitrance of the organic substrate and low heterotrophic respiration, due to waterlogging conditions, have been considered as the main factors explaining the usual high SOC contents in mangrove forests. Only recently has attention been directed toward physicochemical stabilization processes (e.g. physical protection, organomineral interactions) which are now gaining recognition[12].

Particulate organic carbon (POC) and mineral-associated organic carbon (MAOC) are two key fractions that play a significant role in SOC persistence[14]. POC generally consists of larger, less decomposed organic fragments, mainly from plant residues. This fraction is more susceptible to microbial decomposition, resulting in a turnover of a few decades[14]. In contrast, MAOC consists of smaller, more decomposed organic molecules that are tightly bound to soil minerals. This fraction is more stable, with a lifespan ranging from centuries to millennia[14,15], since the mineral association offers protection against microbial degradation[13,16].

Iron (Fe) oxides, are key minerals involved in the stabilization of SOC in coastal wetland soils and marine sediments[17–19]. These reactive Fe phases can form stable covalent bonds between their surface hydroxyls and organic functional groups, protecting SOC from microbial degradation[17]. While in well-drained soils, organic matter typically associates with more crystalline Fe oxides, such as hematite and goethite[20,21], in mangrove soils this association probably occurs through co-precipitation with poorly crystalline Fe oxides, such as ferrihydrite and lepidocrocite. This is due to the redox and acid-base oscillations characteristic of mangrove soils, which favor the formation of poorly crystalline oxides over the crystalline ones[3,22].

The unique geochemical conditions in mangrove soils cause Fe to cycle constantly between different prevailing Fe solid phases, i.e., Fe oxides and pyrite[22] As a result, iron-mediated organomineral interactions (FeOMIs) in these soils form through a different dynamic compared to other waterlogged environments. Considering the drastic geochemical changes associated with LUC in mangroves—such as vegetation removal, drainage, and altered hydrodynamics[23,24]—further research is necessary to understand the precise impact of LUC on FeOMIs and, consequently, on SOC persistence.

Here we evaluate the impact of LUC, specifically the conversion of mangrove forests into pasture and shrimp ponds, on FeOMIs in Amazonian mangroves (Fig. 1). We aimed to provide a comprehensive investigation of the nature and thermal stability of these associations and hypothesized that FeOMIs are critical to SOC persistence but highly susceptible to LUC. To test our hypothesis, we conducted a thorough analysis by combining physical fractionation of SOC, wet-chemical extractions, thermal and spectroscopic analyzes to probe FeOMIs in Amazonian mangrove soils under LUC.

## Results and discussion
### The nature and stability of FeOMI in mangrove soils

To investigate the mechanisms underlying FeOMIs, we analyzed the samples containing the MAOC fraction from pristine mangrove soils at depths of 0–10, 40–50, and 80–100 cm. These samples underwent Fe sequential extraction, with both Fe and OC concentrations measured at each step. Fourier Transform Infrared Spectroscopy (FTIR) and Thermogravimetric-Differential Scanning Calorimetry (TG-DSC) were also conducted between extraction steps to gain deeper insights into the contribution of each Fe fraction to organomineral interactions and SOC stability (see the "Methods" section and Fig S1 for further details). We used the ratio of aliphatic C-H ($2985–2830\,cm^{-1}$) relative to

carboxylate group ($COO^{-};1440–1390\,cm^{-1}$) to assess the degree of organic matter decomposition. A lower ratio suggests increased microbial oxidation, thus increased decomposition[25]. To assess the thermal stability of Fe-bound OC, we calculated energy density (ED) and the temperature at which half the exothermic energy is released ($DSC\text{-}T_{50}$). Energy density can serve as a proxy for the biochemical complexity of organic matter and its degree of decomposition, where higher ED implies a more complex and less decomposed organic structure[26]. Similarly, a higher $DSC\text{-}T_{50}$ indicates greater thermal recalcitrance of SOC[27].

The progressive extraction of Fe was accompanied by a decrease in OC as evidenced by the reduction in the intensities of organic bands in the FTIR spectra (Fig S2a). Simultaneously, there was expansions and shifts to higher temperatures in the exothermic region related to organic matter oxidation (Fig S2b), as shown by increases in both ED and $DSC\text{-}T_{50}$ (Table S1), implying greater thermal stability of the remaining OC. This increase in thermal stability following the removal of Fe initially appears contradictory since Fe-bound OC is commonly perceived as a highly stable fraction of SOC in coastal and marine environments[17,28]. A first explanation for this observation would be the catalytic effect of Fe oxides on organic matter oxidation during thermal analysis[29]. However, when Fe and the associated OC are removed, the remaining OC is still associated with other minerals, such as aluminosilicates, which can also produce a catalytic effect[30]. Thus, an alternative explanation would involve the different OC composition in MAOC subfractions.

The increased thermal resistance of the remaining OC suggests that Fe binds to thermally and (likely) biologically labile OC. This aligns with the "bioenergetic framework"[26,31] and the current paradigm of soil organic matter persistence[13,16,32]. These perspectives suggest that soil organic matter exists as a continuum, processed by the decomposer community from large biopolymers to small monomers. Depolymerization and oxidation of larger organic molecules solubilize smaller ones, which make them more accessible for microbial uptake if the dissolved organic matter is readily available[13]. On the other hand, oxidation enhances chemical reactivity of organic compounds by reducing molecular size, increasing polarity, and thus increasing chemical reactivity[33]. This is crucial for the formation of organomineral interactions, especially those with Fe oxides, which involve ligand exchange reactions with oxygen-bearing organic functional groups[21,34].

When compared to POC, MAOC has lower thermal stability, reflected by lower ED and lower activation energy, the latter positively correlated with $DSC\text{-}T_{50}$[26,31]. This is because MAOC is composed of easily metabolizable organic molecules with low thermodynamic stability[26,31]. Thus, a decrease in thermal stability is expected as SOC decomposition progresses, with larger plant biopolymers being processed into smaller and simple molecules[35]. However, variations in stability within MAOC fractions also exists, depending on its chemical composition[36].

Here we observed a positive correlation between $DSC\text{-}T_{50}$ and the $CH:COO^{-}$ ratio (Fig. 2a), meaning that as the $CH:COO^{-}$ ratio increases (i.e., Fe-bound OC is removed), the remaining MAOC becomes more thermally resistant. Similarly, there was an increase in the C:N ratio with Fe-bound OC removal (Fig. 2b). These patterns indicate that Fe-bound OC is more oxidized and degraded, less thermally stable and less energetic than the residual MAOC. Thus, our data show that Fe phases play a key role in protecting a relatively labile SOC fraction in mangrove soils.

The Fe in organomineral complexes ($Fe_{cit}$) was the most abundant fraction, ranging from 3.8 to 5.1 g kg$^{-1}$across the three studied depths (Table S1). However, the molar OC:Fe ratios were relatively low (< 4.0; Table S1), suggesting that while $Fe_{cit}$ is important for organomineral associations due to its high concentration, it isn't particularly effective at binding OC on a per-unit basis of Fe. The precise mechanism of OC stabilization through association with the $Fe_{cit}$ fraction, whether

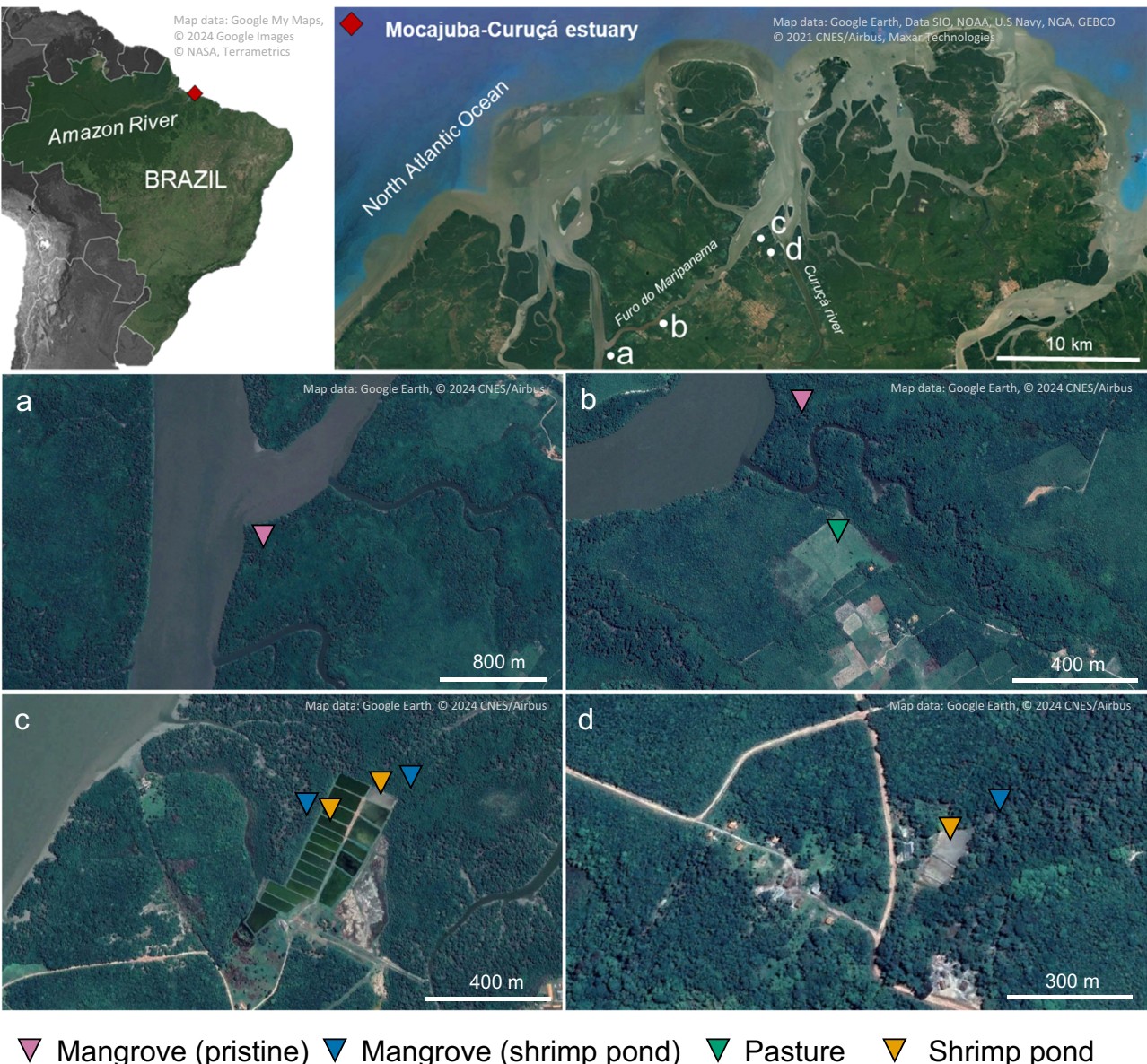

▽ Mangrove (pristine)   ▽ Mangrove (shrimp pond)   ▽ Pasture   ▽ Shrimp pond

**Fig. 1 | Location of the study areas in the Mocajuba-Curuçá estuary within the Amazon region, North Brazil. a** Pristine mangrove, **b** pristine mangrove and the mangrove converted to pasture. **c**, **d** Shrimp ponds and mangroves adjacent to shrimp ponds.

through adsorption or co-precipitation, is uncertain. The stabilization could occur either by cation-bridging or co-precipitation with low molecular weight organic compounds[37], resulting in the observed low OC:Fe ratio.

There was no evident difference in poorly crystalline ($Fe_{asc}$) and crystalline oxides content ($Fe_{cbd}$), with both ranging between 0.5 and 2.0 g kg⁻¹ (Table S1). However, the $Fe_{asc}$ presented the highest OC:Fe ratio ( > 13), nearly 3 times higher than those of $Fe_{cbd}$. While the OC:Fe ratio alone may not offer definitive information about the types of interactions between Fe oxides and OC— since some reducible Fe may not be associated with OC[17]—it does suggest that poorly crystalline oxides exhibit a stronger affinity for OC. Indeed, the adsorption capacity of OC on poorly crystalline Fe oxides can be up to three times higher than on crystalline ones, under identical chemical conditions (Table S2), largely due to their higher surface area and smaller particle size[38]. Additionally, interactions between OC and Fe in poorly crystalline Fe oxides also involve co-precipitation processes[20,21], which can immobilize 10 times more OC than adsorption reactions (Table S2).

Therefore, our data highlight the critical role of Fe oxides, particularly poorly crystalline phases, in protecting OC in mangrove soils.

### Impact of land use change on Fe and SOC biogeochemistry

Conversion of mangroves to pasture and shrimp ponds led to significant changes in the soil geochemical environment as evidenced by the shifts in Eh-pH conditions (Fig. 3). Soil samples from the pristine mangrove and the mangrove adjacent to shrimp ponds were within anoxic to suboxic conditions (Eh -150 ± 96 and -239 ± 89 mV, respectively) and slightly acidic to neutral pH levels (5.8 ± 0.4 and 6.4 ± 0.6, respectively). Such conditions evidence a common soil geochemical environment in mangrove soils favorable to the coupling of iron and sulfate reduction and precipitation of poorly crystalline Fe oxides (see Ferreira et al., 2021). In contrast, the mangrove converted to shrimp pond trended towards more oxidizing conditions (Eh -59 ± 126 mV), mostly within the stability field of crystalline Fe oxides. The physico-chemical changes were more pronounced at the site converted to pasture, where the pH dropped to 4.5 ± 0.5 and Eh increased to

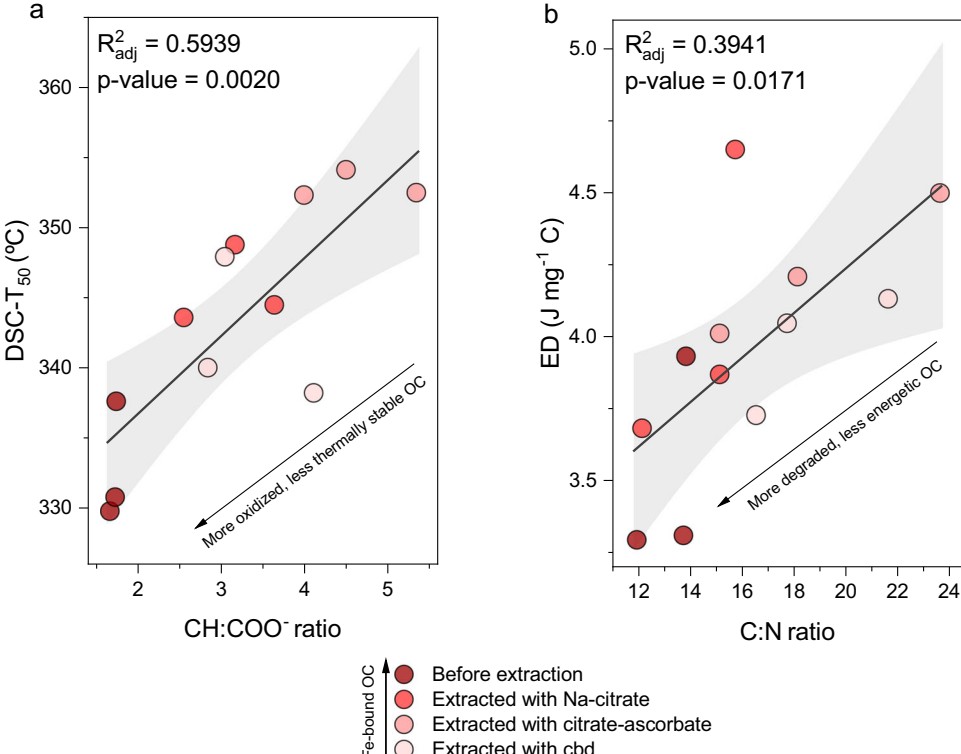

**Fig. 2 | Relationships between thermal properties and chemical composition of organic matter before and after sequential chemical extractions with Na-citrate, citrate-ascorbate, and citrate-bicarbonate-dithionite (cbd). a** Linear regression between DSC-$T_{50}$ and CH:COO⁻ ratio. **b** Linear regression between ED and C:N ratio. DSC-$T_{50}$ and ED represent the temperature at which half of the exothermic energy is released and the energy density of organic matter, respectively, as determined by TG-DSC analysis. The CH: COO⁻ ratio is derived from the ratio of FTIR band areas corresponding to CH and COO⁻ functional groups. Regression lines are shown with 95% confidence intervals (shading).

284 ± 34 mV, indicating oxidizing and acidic conditions and, thus, a sharp shift towards the equilibrium of soluble $Fe^{2+}$.

The observed shifts in Eh and pH were in accordance to changes in Fe geochemistry, which were evident across all soil depths (Fig. 4a). The $Fe_{cit}$ (i.e. Fe fraction in organomineral complexes) was the most sensitive to land use change. Mean concentrations of $Fe_{cit}$ considering all soil depths decreased progressively from 2.9 ± 1.4 g kg⁻¹ in pristine mangroves to 0.7 ± 0.1 g kg⁻¹ in mangroves converted to shrimp ponds and less than 0.2 g kg⁻¹ in those converted to pasture (Fig. 4b). Additionally, there was an important decrease in $Fe_{asc}$ (i.e. poorly crystalline oxides), especially when comparing pristine mangroves ($Fe_{asc}$ = 0.8 ± 0.03 g kg⁻¹) to those converted to pasture (0.3 ± 0.1 g kg⁻¹).

The high variability observed in the shrimp pond samples at each depth are likely due to the drainage and refilling activities associated with shrimp pond management, as well as the death of roots due to deforestation and consequently impacting on soil aeration. These factors likely caused Fe segregation, leading to variability in $Fe_{cbd}$ concentrations across different depths. Despite large variation, there was an eight-fold increase in mean $Fe_{cbd}$ (i.e. crystalline iron oxides) in shrimp ponds in relation to the pristine mangrove and the mangrove adjacent to shrimp ponds (Fig. 4b). This increase is in accordance with the physicochemical conditions (i.e. Eh and pH; Fig. 3) prevailing in the shrimp ponds. Such changes occurred mostly at the expense of the $Fe_{cit}$ fraction (Fig. 4b).

Finally, the total extracted Fe (i.e, the sum of $Fe_{cit}$, $Fe_{asc}$, $Fe_{cbd}$) under pasture was at least five times lower compared to pristine mangroves, with most losses occurring in both $Fe_{cit}$ and $Fe_{asc}$, with a decrease of more than 75% in both fractions (Fig. 4b).

The conversion of mangroves to shrimp pond and pasture led to substantial loss of SOC (Fig. 5a). While in pristine mangroves SOC was generally greater than 30 g kg⁻¹, in shrimp ponds and pasture SOC decreased to less than 7 g kg⁻¹. Most SOC in mangroves, whether pristine or in forests adjacent to shrimp ponds, was found in the form of MAOC (Fig. 5b). These MAOC concentrations ranged from 17 to 25 g C kg⁻¹ being two-fold higher than those of POC, except in the pasture soil (Fig. 5b). Conversion of mangroves into shrimp ponds led to a substantial loss of both MAOC and POC fractions, reaching levels four times lower than the pristine mangroves (Fig. 5b). The conversion to pasture resulted in more pronounced losses, mostly in the MAOC fraction (< 2 g C kg⁻¹; Fig. 5b), indicating that even this presumably stable fraction can be significantly degraded following land use change. Although data on mangrove conversion to pasture in the Amazon is still limited, the observed relative loss of SOC following mangrove conversion to pasture aligns with findings from other regions, where SOC losses are in the range of 80−90%[39,40].

In pristine mangroves, Fe-bound OC represented 8.0 ± 2.0% of the SOC (Fig. 5c). However, this fraction proved highly susceptible to land use change, as evidenced by a decrease of 80% in Fe-bound OC observed across all soil depths in the different LUC scenarios (Fig. 5b). Following LUC, the Fe-bound OC (% SOC) varied between 5.3 ± 5.1% for the mangroves converted to shrimp ponds, and 1.0 ± 1.1% for the mangrove forest converted to pasture (Fig. 5c).

Deforestation followed by soil drainage led to oxidizing conditions and were likely the primary drivers of organic matter decomposition and Fe geochemical changes following conversion to shrimp pond and pasture. A first SOC destabilization process involved the oxidation of organic matter in organomineral complexes containing Fe ($Fe_{cit}$). Thus, as organic matter decomposition proceeded in the converted areas, $Fe_{cit}$ was progressively precipitated from the soil solution as crystalline Fe oxides (e.g. conversion to shrimp pond) or lost as $Fe^{2+}$ with water transport (e.g. conversion to pasture).

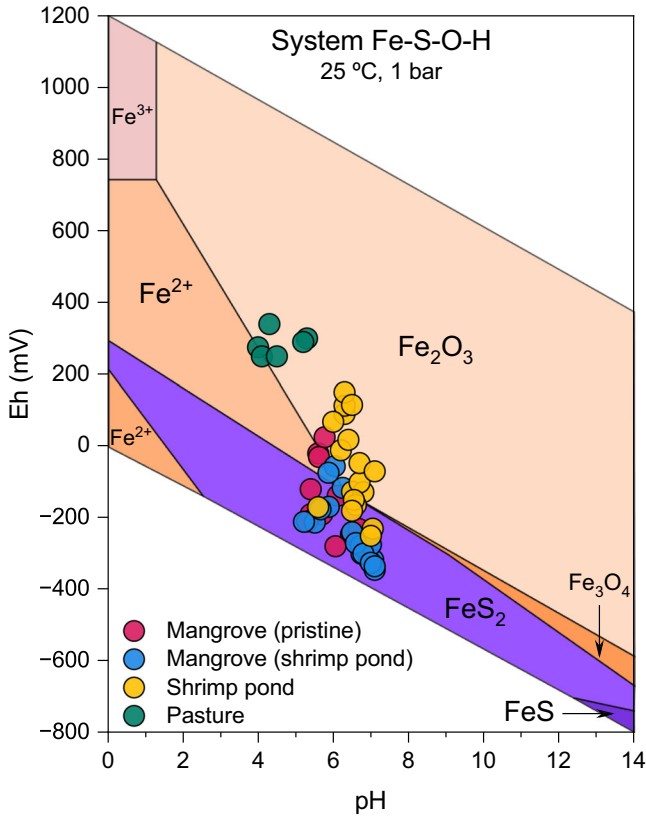

**Fig. 3 | Soil geochemical parameters from mangroves, shrimp pond and pasture samples.** Eh-pH diagram illustrating the stability fields for the Fe-S-O-H system[62] for the studied samples across all depths (0–10, 10–20, 30–40, 40–50, and 80–100 cm).

Another critical process that triggered Fe geochemical changes was the oxidation of pyrite. There was a sharp decrease in pyritic Fe contents from pristine mangroves (mean pyritic Fe = 3.8 g kg$^{-1}$) to shrimp ponds (0.7 g kg$^{-1}$) and pasture (< 0.01 g kg$^{-1}$; Fig S3). Pyrite is an easily weatherable mineral under the surface oxidizing conditions and an important precursor for Fe oxides formation in mangrove and other wetland soils[41], as expressed in Eq. 1 and 2:

$$FeS_2 + H_2O + \frac{7}{2}O_2 \rightarrow Fe^{2+} + 2SO_4^{2-} + 2H^+ \tag{1}$$

$$2Fe^{2+} + \frac{1}{2}O_2 + 3H_2O + \rightarrow 2FeOOH + 4H^+ \tag{2}$$

Typically, as $O_2$ is introduced into soil solution, the reaction produces $Fe^{2+}$ which is rapidly oxidized, resulting in the preferential neoformation of poorly crystalline oxides rather than crystalline ones[42]. This was observed in the mangrove soils where the high amounts of organic matter inhibit the formation of crystalline Fe oxides[42,43], explaining the higher concentrations of Fe$_{asc}$ (Fig. 4). On the other hand, upon mangrove removal, followed by SOC loss through decomposition, crystal nucleation and growth were probably facilitated, favoring the precipitation of $Fe^{2+}$ as crystalline oxides, likely goethite[41,42]. This process also explains the observed increase in the Fe$_{cbd}$ fraction in shrimp ponds (Fig. 4) where the geochemical conditions indicate favorable conditions to both the formation and stability of crystalline Fe oxides (i.e., pH values close to neutrality and Eh values > + 100 mV)[38]. However, the formation of crystalline Fe oxides did not reverse the loss of Fe-bound OC (Fig. 5c), since these oxides are

not as effective as organically complexed Fe and poorly crystalline oxides to immobilize organic matter[21].

A similar outcome would, thus, be expected upon the conversion of mangrove forests to pasture; however, the highly oxidizing conditions resulted in further pyrite oxidation (Fig S3), leading to substantial soil acidification (Eq. 1 and 2). The soil pH of ~4.5 in the pasture (Fig. 3) is a record of the acid dissolution of poorly crystalline Fe oxides[44] which led to $Fe^{2+}$ loss with water transport. Although the pH measured was 4.5 at the time of sampling, it is possible that the pH was even lower during periods of pyrite oxidation, leading to more acidic conditions and, consequently, explaining the massive Fe losses (Fig.4b).

The shifts in Fe oxides crystallinity, upon conversion to shrimp ponds, and the massive Fe losses, upon conversion to pasture, resulted in the destabilization of Fe-bound OC as less crystalline phases were either dissolved or transformed into more crystalline ones, which are less efficient in retaining SOC (Fig. 6).

## Implications for future restoration efforts

Our data reveal a dynamic equilibrium between Fe and SOC that maintains a significant pool of a priori labile SOC fraction in pristine mangroves (approximately 8% of total SOC), which is highly susceptible to the geochemical changes promoted by LUC (Fig. 6). These processes have major implications for the restoration and carbon capture in mangrove forests under land use change.

The substantial Fe loss poses a challenge for restoring the overall biogeochemical functioning of these ecosystems. Despite efforts to accelerate forest recovery through reforestation, the re-establishment of the geochemical balance in mangrove soils may be a gradual process. The Fe dynamics in mangrove soils is influenced by the amount and quality of sediments deposited in estuaries[22]. While mangroves in the Amazon experience relatively high sedimentation rates of approximately $1.5 \pm 0.3$ cm yr$^{-1}$ cm[45], it strongly outcompetes with equally high erosion rates resulting from anthropogenic impact[46]. Consequently, the natural replenishment of Fe in mangrove soils is likely to occur in the long-term.

Thus, current restoration approaches, focusing solely on reforestation, may not sufficiently address the geochemical restoration required to reestablish FeOMIs. Given the critical role of these interactions in protecting SOC, we advocate for broader restoration strategies that incorporate not only vegetation management but also interventions aimed at restoring the soil geochemical environment.

## Methods
### Study area and sampling
The study area is in North Brazil (Pará State) at the Mocajuba-Curuçá estuary (0°42'56"S, 47°54'1"W), east of the mouth of the Amazon River (Fig. 1). The regional climate is tropical monsoon (Am), according to the Köppen classification, with annual rainfall > 3000 mm (Andrade et al., 2017). Local mangroves stands average 20 m in height, largely dominated by red (*Rhizophora mangle*) and black (*Avicennia germinans*) mangroves and subjected to a tidal range of 0.5–5 m. In the past decades, these mangroves have been experiencing anthropogenic pressure from shrimp farming and conversion to pastures[8].

The sampling plots were distributed along the margin of the Furo do Maripanema river, encompassing pristine control mangrove forests and areas converted to shrimp ponds and pasture (Fig. 1). A total of 9 soil cores were collected, including samples from two pristine mangroves, three mangroves adjacent to shrimp ponds, three shrimp ponds, and one pasture (Fig. 1a and b). In each sampling plot, an undisturbed soil core was collected using a stainless-steel sampler for waterlogged soils. Samples were taken at the depths of 0–10, 10–20, 20–30, 30–50, and 80–100 cm. Oxygen-reduction potential (ORP) and pH were measured in the soil samples immediately after collection, using a HANNA HI98121 meter. Redox potential (Eh) was determined

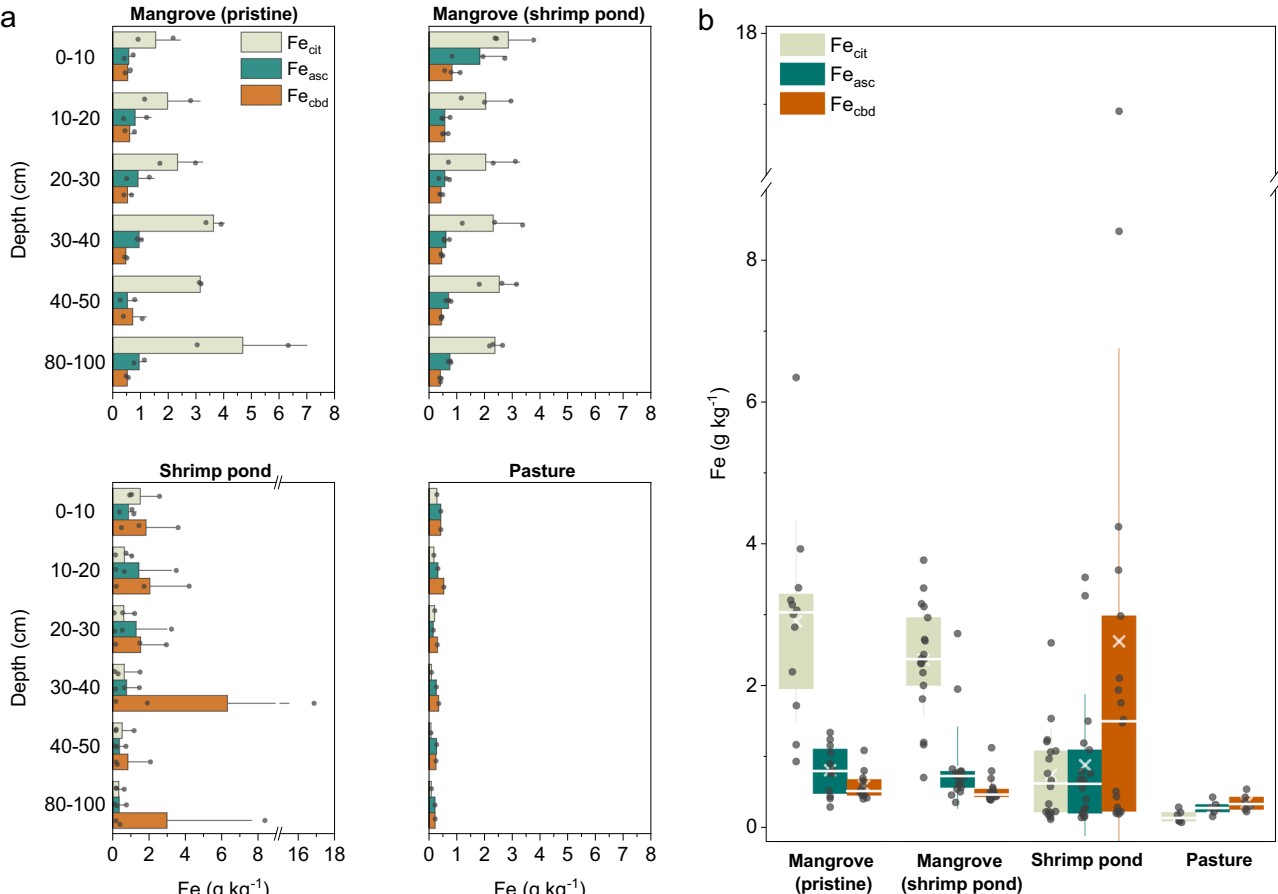

**Fig. 4 | Iron (Fe) concentrations obtained from the sequential wet-chemical extractions. a** Iron extracted with sodium citrate ($Fe_{cit}$), citrate ascorbate ($Fe_{asc}$), and citrate-bicarbonate-dithionite ($Fe_{cbd}$) solutions, across all soil depths. **b** $Fe_{cit}$, $Fe_{asc}$, $Fe_{cbd}$ including all depths from 0 to 100 cm. In the box plots, whiskers represent standard deviations, the box shows the interquartile range (25th to 75th percentile), the line inside the box indicates the median, and "x" denotes the mean.

based on the reference calomel electrode, adjusted from the ORP platinum electrode by adding +244 mV.

Thereafter, the samples were carefully placed in hermetic sealed plastic bags and transported in thermal containers to ensure their preservation until storage at −20 °C. Subsequently, the frozen samples were transported to the laboratory, where they were stored at −20 °C until subjected to analyzes.

## Physical fractionation of SOM
Size fractionation was carried out following a previous method[47]. Initially, wet soil samples equivalent to 20 g of dry mass were added to centrifuge tubes with 80 ml of deionized water. The vials underwent shaking for 1 hour at 120 rpm, centrifugation for 15 min at $3000 \times g$ and the supernatant was discarded. This step aimed to eliminate soluble salts that could interfere with physical dispersion. Subsequently, the sample was transferred to glass vials, filled with 60 ml of deionized water, and ultrasonic dispersion was performed with an energy input of 350 J ml⁻¹. The vials were surrounded by ice-cold water to avoid overheating during sonication. The resulting suspension was wet sieved through 2000–250, 250–53, and 53–20 μm. The particles retained on the sieves (i.e., all particles > 20 μm) contained the particulate organic carbon (POC) fraction. These materials were collected, oven dried at 60 °C, ground and sieved (<150 μm) before organic carbon determination. The suspension containing the particles that passed through the 20 μm sieve were collected in a glass bowl, transferred to centrifuge tubes and subjected to centrifugation ($6000 \times g$ for 10 min) to isolate the <20 μm fraction, which contained the mineral-associated organic C (MAOC). This fraction was freeze-dried,

ground, and sieved (<150 μm) before analyzes. Since none of the samples showed a reaction with 10% HCl, indicating no need for pre-treatment, they were directly subjected to elemental analyzes. All fractions, including bulk soil samples prior to fractionation, were analyzed for OC content using a Leco TruSpec CHN combustion elemental analyzer.

It is worth noting that this size fractionation method, despite the potential recovery of some fine POC in the <20 μm fraction, is widely accepted for effectively separating POC and MAOC, offering assertive interpretations regarding the ecological functioning of these operational fractions[14,48,49]. Importantly, this methodology ensures minimal impact on the sample's chemical composition as no chemicals were added, thus preserving the nature of the organo-mineral associations.

## Wet-chemical extractions for assessing FeOMI
The size fraction containing the MAOC underwent a sequential wet-chemical extraction for quantifying and characterizing FeOMI. This method is an adaptation from that developed by Lalonde et al., (2012).consists of sequential extractions with Na-citrate ($Fe_{cit}$), citrate-ascorbate ($Fe_{asc}$) and citrate-bicarbonate dithionite ($Fe_{cbd}$).

We added 40 ml of solution to 0.50 g of freeze-dried and ground samples in 50 ml centrifuge tubes. The first extraction step was carried out with 0.2 M Na-citrate buffered with citric acid at pH 6.0 (Fecit). In this step, passive Fe is extracted by complexation with citrate, an efficient complexing agent that partially dissolves Fe from the surfaces of poorly crystalline Fe oxides[44] and Fe from organo-mineral complexes[50,51].

Subsequently, the residue underwent extraction with 0.2 M sodium citrate-0.05 M sodium ascorbate at pH 6.0 ($Fe_{asc}$). This step

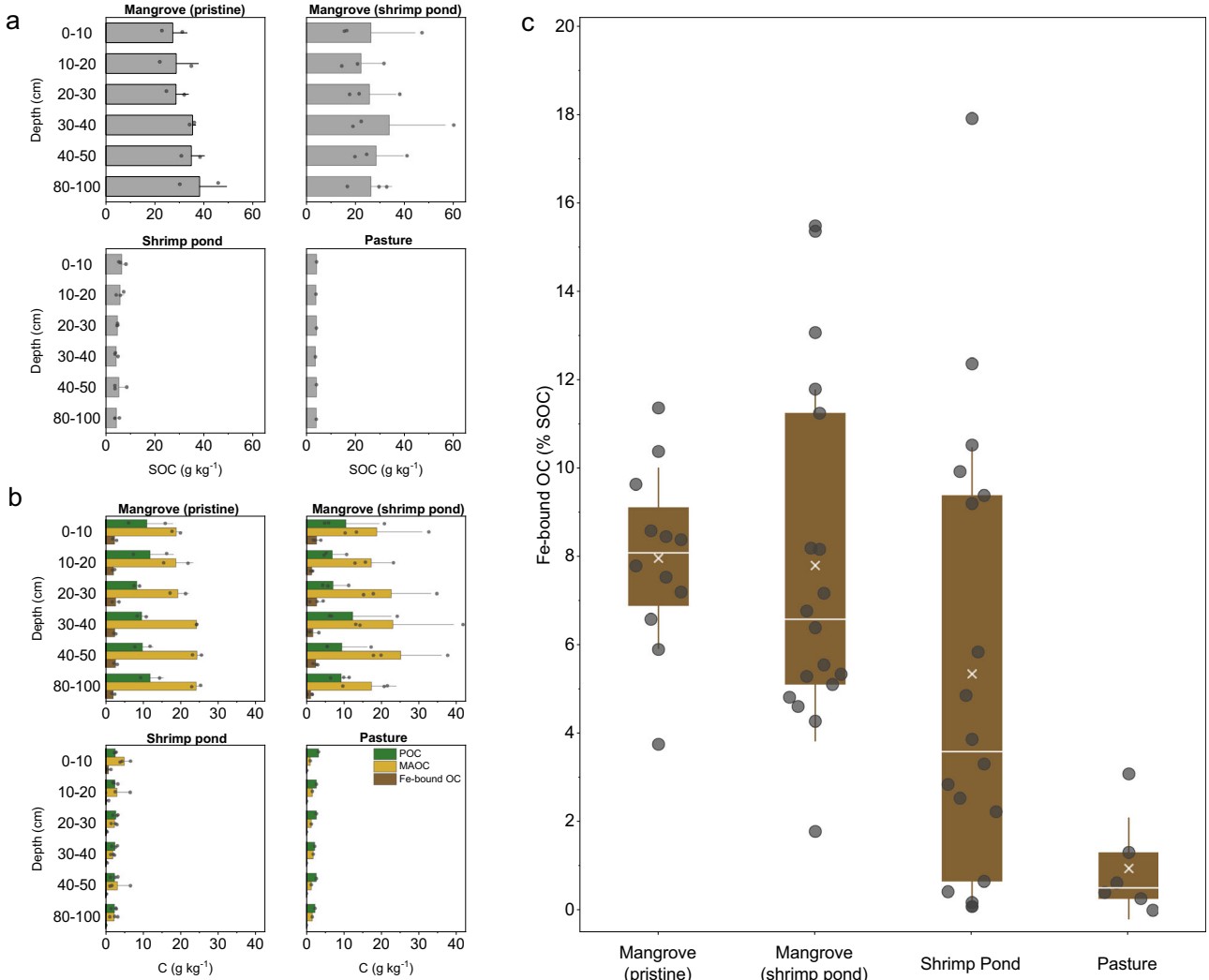

**Fig. 5 | Organic carbon concentration in the different operational fractions.**
**a** Total soil organic carbon (SOC) **b** Particulate organic carbon (POC), mineral-associated organic carbon (MAOC) and iron bound organic carbon (Fe-bound OC) concentrations across all soil depths and LUC scenarios. **c** Fe-bound OC expressed as % of SOC, including all soil depths from 0 to 100 cm. In the box plots, whiskers represent standard deviations, the box shows the interquartile range (25th to 75th percentile), the line inside the box indicates the median, and "x" denotes the mean.

maintained citrate as a complexing agent and introduced ascorbic acid as a mildly reducing agent. Ascorbic acid reduces poorly crystalline Fe oxides but not crystalline ones, making this extraction specific to poorly crystalline Fe oxides such as ferrihydrite and lepidocrocite[51].

For the $Fe_{cbd}$, the residue reacted with 0.27 M trisodium citrate, 0.11 M sodium bicarbonate, and 0.1 M sodium dithionite without heating[52]. This step was aimed at extracting the remaining Fe oxides, focusing specifically on crystalline Fe oxides such as hematite and goethite.

Control samples were extracted with NaCl at ionic strength equivalent to the "cbd" extraction as organic matter held in the exchange phase could desorb due to ionic strength effects[19].

The extractions involved shaking at 160 rpm for 16 h at room temperature, with the residue rinsed with 30 ml of deionized water between each step. Supernatants were separated from the solid fraction through centrifugation at $3000 \times g$ for 10 min and filtered through 0.45 µm glass fiber filters. The solid residue was then rinsed three times with 0.1 M NaCl to eliminate any residual organic reagent that could interfere with further analysis. The Fe content in the extracts was determined using an atomic absorption spectrometer (PerkinElmer 1100B), while the concentration of organic carbon associated with Fe (Fe-bound OC) was calculated accounting for mass losses after the

sequential wet-chemical extractions, as in Eq. 3:

$$Fe-bound\ OC = \frac{OC_i \times weight_i - OC_f \times weight_f}{OC_i \times weight_i} \times 1,000 \quad (3)$$

Where Fe-bound OC is expressed in g kg$^{-1}$; $OC_i$ and $OC_f$ are the OC concentration, in %, in the sample before and after the extraction; $weight_i$ and $weight_f$ are sample weight, in g, before and after the extraction. The results were subtracted from that of the controls to yield the final Fe-bound OC values. We also expressed Fe-bound OC as the relative content in SOC (% SOC) and the ratio between OC and Fe extracted with cbd were expressed as OC:Fe in mol mol$^{-1}$.

To assess the impact of freeze-drying on potential changes in Fe oxide crystallinity and their extractability, we conducted a control experiment in which both freeze-dried and moist samples were subjected to the same extractions. The same extractions were performed on bulk samples (i.e., without any physical fractionation). There were only small differences in Fe extraction yields, with notable variations observed only in the pasture samples, where Fe-citrate and Fe-citrate-bicarbonate-dithionite were generally higher in moist samples. However, the proportion of Fe extracted across fractions between freeze-dried and moist samples remained very similar. Thus, no significant

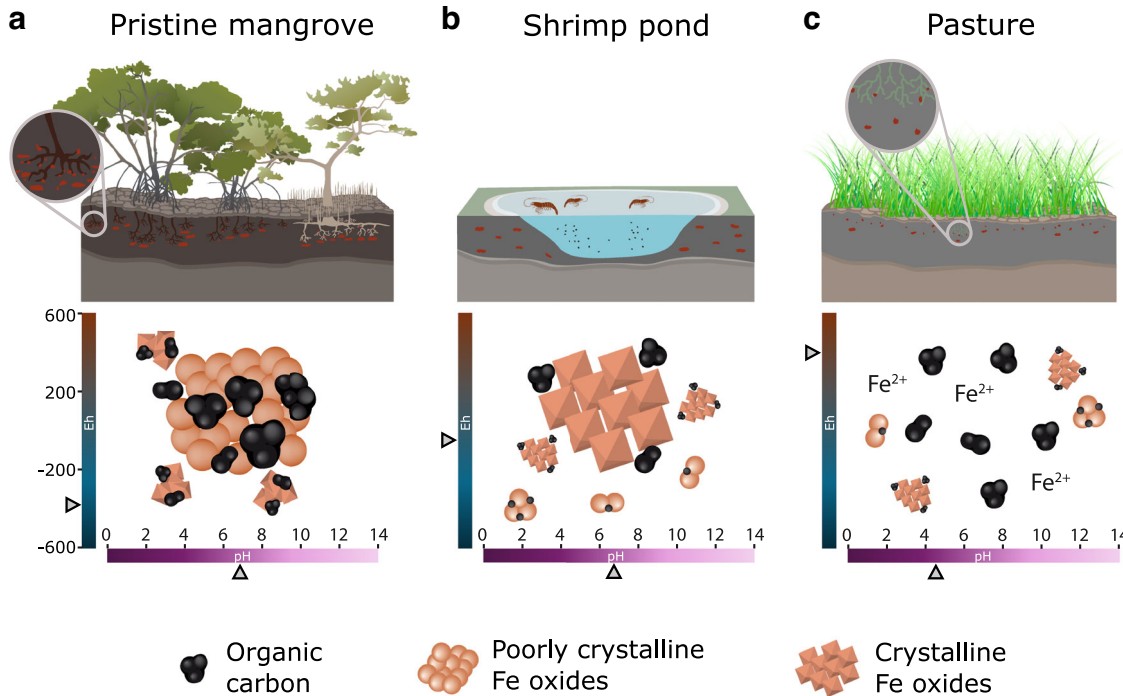

**a** Pristine mangrove    **b** Shrimp pond    **c** Pasture

● Organic carbon    ⬤ Poorly crystalline Fe oxides    ◆ Crystalline Fe oxides

**Fig. 6 | Changes in iron-mediated organo-mineral interactions (FeOMIs) following land use change. a** In pristine mangroves, FeOMIs are primarily governed by the association between organic matter and poorly crystalline iron oxides, mainly through co-precipitation, which effectively immobilizes organic carbon. **b** As mangroves are converted to shrimp ponds the more oxidative environment leads to an increase in crystalline iron oxides. These crystalline forms of iron oxides are less efficient at immobilizing organic matter, as they predominantly rely on adsorption. **c** Following conversion to pasture, the geochemical environment undergoes strong oxidation coupled to acidification. This causes the dissolution of iron oxides, leading to the massive release of Fe-bound OC. Figure adapted from BioRender (Gomes Viana, D. 2024 https://BioRender.com/m14f407). and adapted and modified from Integration and Application Network (ian.umces.edu/media-library), licensed under Attribution-ShareAlike 4.0 International (CC BY-SA 4.0) https://creativecommons.org/licenses/by-sa/4.0/. This adapted figure is distributed under the same license (CC BY-SA 4.0).

effect of Fe remobilization was detected across the different Fe fractions. See Tables S3 and S4 for further details.

Another experiment was conducted using subsamples of the MAOC fraction from one of the studied pristine mangroves (00°45'24"S 47°57'19"W) at the depths of 0−10, 40−50 and 80−100 cm, in duplicate, to further investigate the mechanism underlying FeOMI. In this case, the subsamples were analyzed after each step of the sequential extraction (see the scheme in Fig S1). Besides the determination of Fe concentration and Fe-bound OC, the samples were also analyzed for total nitrogen TN. Additionally, composite samples (from the duplicates) underwent infrared and thermal analyzes.

### Wet chemical extractions in bulk samples

To estimate the concentrations of reactive Fe−iron phases that are reductively dissolved by sodium dithionite[19]− and pyritic Fe, we performed a sequential wet chemical extraction on moist bulk samples. The method is adapted from previous methods[53,54]. The procedure involved adding 30 ml of 0.25 mol L$^{-1}$ sodium citrate + 0.11 mol L$^{-1}$ of sodium bicarbonate and 1 g of sodium dithionite to 2 g of sample. The mixture was stirred for 30 min at 75 °C to extract the reactive Fe. Following this, silicate-associated Fe was removed by stirring the sample with 30 mL of 10 mol L$^{-1}$ HF for 16 hours, and the organic phase was removed by stirring with 10 mL of concentrated $H_2SO_4$ for 2 hours. Finally, the pyritic Fe was extracted by stirring the samples with 10 mL of concentrated $HNO_3$ for 2 h at room temperature. The samples were rinsed between each extraction, and the supernatant was collected for Fe determination using atomic absorption spectroscopy (AAS).

### Infrared spectroscopy

The samples used in the additional sequential extraction experiment were analyzed through transmission Fourier transform infrared spectroscopy (FTIR) using a Shimadzu FTIR Prestige-21 with a DLATGS detector. The sample was analyzed as KBr pellets and spectra were obtained from 4000 to 400 cm$^{-1}$ at a resolution of 1 cm$^{-1}$. Each sample was scanned 64 times and averaged to generate the final spectrum. The spectra were baseline corrected, normalized (min-max) and peak area of the regions of interest were calculated without smoothing. The spectra were subsequently smoothed for improved visual presentation using the Savitzky-Golay method with a quadratic function and 11 data points. All these procedures were done using the Orange 3.36 software. Organic functional group assignments were based on the literature[55].

We used the ratio of aliphatic C-H (2985-2830 cm$^{-1}$) relative to COO$^-$ (1440−1390 cm$^{-1}$) to assess the degree of organic matter decomposition. A decrease in this ratio may reflect increased microbial oxidation and/or decreased plant contribution. This approach follows the same rationale as that of[25,56,57]. However, we used the COO$^-$ (1440−1390 cm$^{-1}$) instead of C = O (1760−1550 cm$^{-1}$), as the latter overlaps with alkenyl (1680−1620 cm$^{-1}$) and aromatic ring (1615−1580 cm$^{-1}$) stretches and the region is strongly influenced by mineral interference[58], especially considering that we analyzed the MAOC fraction.

### Thermal analysis

The same samples used in FTIR analysis were subjected to Thermogravimetric analysis (TG) and differential scanning calorimetry (DSC) to evaluate the thermal stability of MAOC and Fe-bound OC. The thermograms were collected using a Discovery SDT 650 analyzer (TA instruments), under a dry synthetic-air atmosphere consisting of 80% $N_2$ + 20% $O_2$, with a flow rate of 50 mL min−1. The samples were heated from 25 to 100 °C, held isothermally until mass equilibration, then heated to 600 °C at a rate of 10 °C min$^{-1}$. This approach is often used as

a proxy to assess the stability of the organic matter associated with minerals[59,60].

An inflection in the raw DSC thermogram was observed near 210 °C, suggesting a shift from endothermic water loss to exothermic organic matter oxidation[27]. Additionally, the strong inflection around 480 °C indicated the transition from organic matter oxidation to the dihydroxylation of kaolinite (the main clay mineral in the studied soils as confirmed by FTIR analysis, see Fig S4), which occurs around 500 °C[61]. Thus, these two regions were set as baseline points for organic matter oxidation.

We integrated the DSC heat flux, in mW mg$^{-1}$, to obtain the total exothermic content, in mJ mg$^{-1}$, over the exothermic region 210-480 °C. From the total exothermic content, two parameters were derived: the temperature at which half of the total exothermic energy had been released (DSC-T$_{50}$) and the energy density (ED), in J mg$^{-1}$ OC. The latter was obtained by dividing the total exothermic content by the OC concentration in (%). All the procedures were made using the Origin 2022b software.

## Statistical analysis

To analyze the variation in measured parameters, including iron and carbon fractions, Eh, and pH, we applied Permutational Multivariate Analysis of Variance (PERMANOVA) using the adonis2 function from the vegan package in R. Euclidean distance matrices were computed for each variable, and PERMANOVA was performed to evaluate the effects of site (pristine mangroves, mangroves adjacent to shrimp ponds, shrimp ponds, and pasture) and depth. Site was treated as a fixed factor, while depth was nested within site. The analysis was based on 999 permutations, followed by pairwise comparisons between sites with Bonferroni correction for multiple comparisons. The complete results are available in the "Extended_data" file. Additionally, we performed linear regressions to explore relationships between parameters obtained from TG-DSC, FTIR analyzes, and C:N ratio.

## Data availability

The data that support the findings of this study are available in figshare with the identifier https://doi.org/10.6084/m9.figshare.27111262.

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

## Acknowledgements

We gratefully acknowledge the support of the RCGI—Research Center for Greenhouse Gas Innovation (23.1.8493.1.9), hosted by the University of São Paulo (USP) and sponsored by FAPESP—São Paulo Research Foundation (20/15230-5), and PETRONAS Petróleo Brasil Ltda. The authors acknowledge the strategic importance of the support given by ANP (Brazil's National Oil, Natural Gas and Biofuels Agency) through the R&DI levy regulation (ANP–Project #23.702-4; BlueShore). We appreciate the support from the Center for Carbon Research in Tropical Agriculture (CCARBON) at the University of São Paulo, sponsored by the São Paulo Research Foundation (FAPESP) under grant 21/10573-4. This study was further supported by grant 23/06841-9, São Paulo Research Foundation (FAPESP) to F.R., and grant 305013/2022–0, National Council for Scientific and Technology Development (CNPq) to T.O.F. Additionally, A.F.B. was supported by the National Geographic Society and Rolex Perpetual Planet Amazon Expedition (PFA-21-PP031), C.R. was supported by the Fair Carbon project TROPECOS (ANR-22-PEXF-012) and X.L.O. was supported by the Consellería de Educación, Universidade e Formación Profesional-Xunta de Galicia (Axudas á consolidación e estruturación de unidades de investigación competitivas do SUG do Plan Galego IDT, Ambiosol Group ref. ED431C; 2022/40. We also thank our lab technician, Leandro Luis Goia, for his valuable assistance with the wet-chemical extraction analyses.

## Author contributions

F.R.: Conceptualization, Methodology, Formal analysis, Investigation, Writing–Original Draft, Writing- Review & Editing, Visualization. A.F.B.: Conceptualization, Investigation, Resources, Writing—Review & Editing, Project administration, Funding acquisition. H.M.Q.: Formal analysis, Investigation, Writing—Review & Editing, Visualization. X.L.O.: Writing - Review & Editing, Investigation, Visualization. C.R.: Writing—Review & Editing, Investigation. T.O.F.: Conceptualization, Writing- Review & Editing, Resources, Funding acquisition.

## Competing interests

The authors declare no competing interests.
