## [Peer Review File · Nature Communications]

Iron's role in soil organic carbon (de)stabilization in mangroves under land use change

Corresponding Author: Professor Tiago Ferreira

Version 0:

Reviewer comments:

Reviewer #1

(Remarks to the Author)
Review of submitted paper

Title: Iron's role in soil organic carbon (de)stabilization in mangroves under land use change
Authors: F. Ruiz, A. Fraga Bernardino, H. Melo Queiroz, X. Luis Otero, C. Rumpel, T. Osório Ferreira

General comments

The authors submitted a manuscript describing the role of iron in the cycling of organic carbon in Brazilian mangroves and the effect of land use change (LUC) on these interactions. They find that LUC leads to large differences in Fe concentrations, in the crystallinity of Fe minerals, and in the amount of OC associated with the different fractions of Fe amorphous or crystalline precipitates.

This is a solid overall paper, well written and easy to understand even for the non-specialist. I do not have any major concerns therefore this review is short and amounts to a series of minor comments and requests for changes detailed in the Specific Comments section below. The data appears solid, although I feel that the reason(s) for some of the most variable results (high standard deviations) should be explained. The topic is interesting and the experimental design solid, although I would have liked to see more than just one pasture sample being analyzed (the authors should comment on the representativity of this single core with respect to other lands converted from mangrove to pasture).

The only important concern that I have is linked to the sample preparation prior to the sequential extraction scheme: freeze-drying is known to significantly affect the crystallinity of amorphous and poorly crystalline minerals. The authors have not addressed, and should be addressing this point as explained in comment #21 below.

I recommend publishing this manuscript but only after minor corrections.

Specific comments

1. Abstract, line 7: The result mentioned here is not contrary to prevailing assumptions.
2. Lines 105-111: The authors should briefly mention the advantages and disadvantages of using thermal resistance as a proxy for biological lability in addition to referring to the papers of Plante and al.
3. Lines 116-119: The increase in stability makes total sense if we assume, as the authors do, that the Fe-associated OM fraction is more labile than the non-Fe-associated fraction, as has been shown several times in the past. If the more labile Fe-associated OM is removed from the total sample, then the residual, non-Fe-associated fraction will appear more refractory, as seen here (and as suggested on lines 129-131).
4. Lines 134: What "strong correlation"? Figure S2 shows the graphical results from a PCA analysis. Please explain.
5. Lines 148-151: The OC:Fe ratio is not a reliable clue for the type of interactions between iron (oxyhydr)oxides and OC, as explained in Barber et al. (2017), because a fraction of reducible Fe is found unassociated with OC.
6. Lines 158-165: This is an elegant way of demonstrating what was already assumed (that OC associated predominantly with amorphous iron (oxyhydr)oxides).
7. Lines 174-176: There seems to be a slight shift towards more oxidizing conditions indeed, but is this shift significant?

Please provide the statistical support for this claim (or lack of support). Given the spread of the data, their overlap also, it is difficult to say whether this shift is statistically significant.

8. Line 187: What do these average and standard deviation correspond to? All datapoints for the pristine mangrove and mangrove adjacent to the shrimp ponds? Please explain.

9. Lines 187 and 188: Please provide an explanation and a standard deviation for the values for the shrimp ponds and the pasture.

10. Lines 190-191: The first two values are not significantly different given the very large standard deviation for the shrimp ponds. Also, please provide a standard deviation for the pasture sample (again, it is a depth-average average, not a single datapoint).

11. Figure 4a: How do you explain the very high standard deviations found for the shrimp pond samples at each depth? Also, how do you explain the variations between different depths, also for the shrimp pond? You report a 6-fold decrease in Fe-citrate concentration between the 30-40 cm layer and the 40-50 cm layer – a huge decrease for two adjacent layers.

12. Figure 4b: Please add the standard deviations for the pasture sample since the values are depth-averaged.

13. Lines 207-208: Except for the pasture sample – please specify.

14. Line 219: Again, please add the standard deviation for the pasture sample here and in Figure 5c.

15. Lines 259-260: This is entirely speculative as there is no data in this work that suggests oxidation of pyrite, aside from a decrease in pH that could have different causes. Please indicate that this is a speculative explanation in the text.

16. Line 260: The soil pH is between 4 and 6, with an average of about 5, which is one order of magnitude higher in H⁺ concentration compared to what is written in the paper (which is misleading). On Figure 3, all but one or two datapoints fall into the Fe₂O₃ stability zone rather than the Fe²⁺ stability zone of the Pourbaix diagram. The pH is probably a little too high for this explanation to be the only reason for the Fe losses. Can you think of an alternative explanation?

17. Line 288: This is a high sedimentation rate! Iron being a predominant element in soils of Amazonia (laterite soils), the particles carried by rivers must be enriched in this element, and therefore replenishing the soils' Fe content following remediation should not be a lengthy process. While the concluding remarks on lines 288-297 are true to some extent, they paint a darker picture than what the reality probably is. Maybe this should be mentioned?

18. Line 304: Please explain what "Am" means.

19. Lines 333-334: This is not clear. The material retained on which one of the sieves? The first one only? Please be more explicit.

20. Lines 337-338: Please provide the details for the centrifugation step.

21. Line 339: Freeze-drying changes the crystallinity of iron (oxyhydr)oxides, and thus its extractability with the different complexation or reducing agents used in this work. Have you run any before/after freeze-drying controls? How do you know then whether the data obtained after freeze-drying the samples is representative of the in situ samples? Please address this point carefully as it could have led to significant changes in the distribution of OC among the different Fe fractions.

22. Lines 352-373: Please provide the volumes used for the extractions. Also, how did you ascertain that the complexation capacity of Na-citrate at 0.2 M was not surpassed since the initial mass of sediment sieved was 20 g? The same is true for the other extractions as well (complexation capacity and reduction capacity for ascorbate and dithionite).

23. Lines 402-411: Please explain how the samples were prepared for FTIR analysis since the absolute peak intensities are compared between samples. Surely there must have been some sort of normalization? Lower intensities could simply mean less abundant organic matter in the OM-KBr mix rather than a lower relative abundance of a given functional group.

24. Table S1, Fe concentration column: The data should have the same number of decimal points throughout. The number in the second decimal point should be placed in subscript to indicate that it is indicative only rather than significant.

25. Figure S2: More explanation about what we are seeing on this PCA graph should be given in the figure caption.

Sincerely,

Reviewer #2

(Remarks to the Author)

The research presented by Ruiz et al. investigates the role of Fe in SOC (De)stabilization in mangrove forests under LUC. The research is of environmental relevance, however does not add significantly to the existing literature. It is a good study, but in my opinion does not meet the novelty criteria to be published in Nature Communications. I wish you all the best to the team and look forward to reading more about this research in future.

Reviewer #3

(Remarks to the Author)

This manuscript investigates the impact of iron-mediated organic-mineral interactions (FeOMI) on the stabilization of soil organic carbon (SOC) in Amazon mangrove soils and explores how land use change (LUC) affects this process. The study comprehensively and systematically analyzes the role of iron in the mineral association of organic carbon from multiple perspectives, including physical fractionation, chemical composition, and thermal stability. This work has a profound guiding significance for the exploration SOC stability in mangroves and the protection of soil carbon under land use change. However, the manuscript still presents numerous issues in terms of overall text composition, figure and table presentation, and results analysis.

My first concern is that the massive loss of SOC due to LUC was not supported by the mechanism of FeOMI, as proposed by the authors. Conversion of mangroves to shrimp pond and pasture resulted in substantial loss of SOC, especially for the MAOC fractions. Although loss of Fe-bound OC might be important in mangrove ecosystem, it's not sufficient to explain the contribution of FeOMI in loss of MAOC. Providing direct evidence that crystalline iron retains less soil organic carbon would strongly support the authors' viewpoint.

I agree with first result regarding the critical role of FeOMI in mangrove soils and the explanation that iron binds more labile carbon. However, the evidence for the conclusion that Fe oxides protects more labile SOC fractions was not sufficient. It is plausible that Fe oxides affinity for more thermally labile and oxidized organic compounds. But it can also not exclude the explanation that catalytic effect of Fe oxides on organic matter oxidation during thermal analysis. In addition, thermally labile OC is not equivalent to labile SOC fraction. It's better to provide direct evidence that how Fe oxides protect POC and MAOC. In the introduction, it is necessary to briefly describe the important role of mineral protection mechanisms in the stabilization of soil organic carbon in mangroves. The relationship between the forms of iron, or their crystallinity, and soil organic carbon should be clearly introduced. Additionally, hypotheses about the impact of land use change on iron and the iron-OM bound relationship should be reviewed with published literature.

Table S1, which is a key result, is somewhat difficult to understand. The parameters displayed in the table need clarification. It appears that some indicators refer to the extracted portion while others refer to the residual portion. This should be more clearly indicated in the table explanation.

The discussion on the environmental implications does not seem to independent part. This part does not present any new findings or conclusions. I do not understand what the authors intend to express about the role of the environment in carbon sequestration in mangroves and the iron-OC binding relationship under LUC. Additionally, it is unclear what is meant by "environment". Supplementing this section with specific facts and relevant data would make it more compelling.

There also some other minor errors in the text and figures, such as the lack of a b annotation in Fig.2, and the Table 1 mentioned in L133 is Table S1. I recommend that the authors thoroughly review the entire text to avoid these errors which easily make reader confusing.

Version 1:

Reviewer comments:

Reviewer #1

(Remarks to the Author)

Review of re-submitted paper

Title: Iron's role in soil organic carbon (de)stabilization in mangroves under land use change

Authors: F. Ruiz, A. Fraga Bernardino, H. Melo Queiroz, X. Luis Otero, C. Rumpel, T. Osório Ferreira

General comments

Thank you for your thorough revision of the manuscript. My main concerns were addressed in detail in this new version. I find the changes made to the article satisfactory and thus recommend its publication in the current form.

Sincerely,

Reviewer #3

(Remarks to the Author)

I appreciate the response to my comments. The manuscript has been largely improved.

Author's response to the reviewers

Reviewer #1:

The data appears solid, although I feel that the reason(s) for some of the most variable results (high standard deviations) should be explained.

Authors' response:

We appreciate the reviewer's observation and the opportunity to clarify this point. The high standard deviations observed in the studied soils can be attributed to natural spatial variability. This is especially true for redox-sensitive elements like Fe, where variability is expected even over small sampling distances, both vertically and horizontally, in mangrove soils (Otero et al., 2006). Mangroves are highly dynamic systems, and several factors—including root density, bioturbation, and tidal oscillation—can increase the variability in the data (Araújo Júnior et al., 2016; Otero et al., 2006).

Importantly, despite the high standard deviations, our statistical analyses revealed significant differences between the mangrove sites and the converted areas. This finding reinforces the robustness of our results, as the reviewer noted. To further support this, we have included the complete PERMANOVA test outputs for all measured variables in the "Extended_data" file.

Reviewer #1:

The topic is interesting and the experimental design solid, although I would have liked to see more than just one pasture sample being analyzed (the authors should comment on the representativity of this single core with respect to other lands converted from mangrove to pasture).

Authors' response:

Thank you for your constructive feedback. We carefully selected this core to ensure it was representative of the broader area of mangrove-converted pastures. Previous work by Bernardino et al. (2024) demonstrated minimal variability in SOC (%) within this pasture area, with SOC values ranging from 0.14 ± 0.04 to 0.32 ± 0.09 within the first meter ($n = 6$). Our measured SOC values fall within this range, indicating that the core we analyzed is indeed representative of the larger pasture area.

Although data on mangrove conversion to pasture in the Amazon is still limited, the observed relative loss of SOC following mangrove conversion to pasture aligns with findings from other regions, where losses are typically in the range of 80-90% (Boone Kauffman et al., 2017; Kauffman et al., 2016). Unfortunately, data on Fe geochemistry and Fe-bound OC is currently lacking, as, to the best of our knowledge, no studies have yet been conducted in this regard. This gap further highlights the originality and importance of the present work.

We included a comment on this in lines 230-236.

Reviewer #1:

The only important concern that I have is linked to the sample preparation prior to the sequential extraction scheme: freeze-drying is known to significantly affect the crystallinity of amorphous and poorly

crystalline minerals. The authors have not addressed, and should be addressing this point as explained in comment #21 below.

Authors' response:

Thank you for highlighting this important issue. We acknowledge the potential impact of freeze-drying on the crystallinity of amorphous and poorly crystalline minerals, which could indeed influence the distribution of Fe fractions and associated organic carbon (OC).

Our method was adapted from Lalonde et al. (2012), which specifically quantifies the OC associated with reactive Fe in marine and coastal sediments. Lalonde's protocol includes a freeze-drying step, thus, by using this method, we ensure that our results are comparable to previously published data, which is crucial for consistency and comparability across studies.

The freeze-drying step was particularly critical in our study because we conducted physical fractionation prior to the wet chemical extractions. Thus, it allowed to weight the different size fraction and correctly quantify the POC and MAOC. Moreover, sample homogenization is essential for the wet chemical extraction steps. Without freeze-drying, the samples would have been difficult to homogenize due to sedimentation during centrifugation, which could lead to uneven distribution of materials with different densities. This would have compromised the accuracy of our fractionation and subsequent analyses.

However, we recognize the reviewer's concern regarding the possible alteration of Fe fractions due to freeze-drying. To address this, we conducted a control experiment comparing moist and freeze-dried bulk samples (without physical fractionation).

There were insignificant differences in Fe extraction yields, with notable variations observed only in the pasture samples, where Fe-citrate and Fe-citrate-bicarbonate-dithionite were generally higher in moist samples. However, the proportion of Fe extracted across fractions between freeze-dried and moist samples remained very similar. Thus, no significant effect of Fe remobilization was detected across the different Fe fractions. Even in the pasture samples, where larger differences were observed, the absolute Fe yield was too low to affect the overall conclusions. See Tables S3 and S4 for further details.

We have included this information in lines 426-435.

Reviewer #1:

1. Abstract, line 7: The result mentioned here is not contrary to prevailing assumptions.

Authors' response:

We thank the reviewer's attention to this detail. To clarify our intended message, we have revised the sentence to better convey our findings.

Lines 6-8: "Our results indicate that Fe protects more labile SOC fractions, otherwise vulnerable to biological degradation".

Reviewer #1: 2. Lines 105-111: The authors should briefly mention the advantages and disadvantages of using thermal resistance as a proxy for biological lability in addition to referring to the papers of Plante and al.

Authors' response:

We appreciate the reviewer's suggestion to discuss the advantages and disadvantages of using thermal resistance as a proxy for biological lability. In the revised manuscript, we have added a more comprehensive discussion that also addresses the inquiry from reviewer #3:

Lines 122-149: "The increased thermal resistance of the remaining OC suggests that Fe binds to thermally and (likely) biologically labile OC. This aligns with the "bioenergetic framework" (Williams et al., 2018; Williams and Plante, 2018) and the current paradigm of soil organic matter persistence (Lehmann et al., 2020; Lehmann and Kleber, 2015; Schmidt et al., 2011). These perspectives suggest that soil organic matter exists as a continuum, processed by the decomposer community from large biopolymers to small monomers. Depolymerization and oxidation of larger organic molecules solubilize smaller ones, which make them more accessible for microbial uptake if the dissolved organic matter is readily available (Lehmann et al., 2020). On the other hand, oxidation enhances chemical reactivity of organic compounds by reducing molecular size, increasing polarity, and thus increasing chemical reactivity (Kleber et al., 2015). This is crucial for the formation of organomineral interactions, especially those with Fe oxides, which involve ligand exchange reactions with oxygen-bearing organic functional groups (Chen et al., 2014; Gu et al., 1995).

When compared to POC, MAOC has lower thermal stability, reflected by lower ED and lower activation energy, the latter positively correlated with DSC-T₅₀ (Williams et al., 2018; Williams and Plante, 2018). This is because MAOC is composed of easily metabolizable organic molecules with low thermodynamic stability (Williams et al., 2018; Williams & Plante, 2018). Thus, a decrease in thermal stability is expected as SOC decomposition progresses, with larger plant biopolymers being processed into smaller and simple molecules (Barré et al., 2016). However, variations in stability within MAOC fractions also exist, depending on its chemical composition (Nguyen et al., 2019).

Here we observed a positive correlation between DSC-T₅₀ and the CH:COO⁻ ratio (Fig 2a), meaning that as the CH:COO⁻ ratio increases (i.e., Fe-bound OC is removed), the remaining MAOC becomes more thermally resistant. Similarly, there was an increase in the C:N ratio with Fe-bound OC removal (Fig 2b). These patterns indicate that Fe-bound OC is more oxidized and degraded, less thermally stable and less energetic than the residual MAOC. Thus, our data show that Fe phases play a key role in protecting a relatively labile SOC fraction in mangrove soils."

We hope this explanation clarifies our rationale.

Reviewer #1:

3. Lines 116-119: The increase in stability makes total sense if we assume, as the authors do, that the Fe-associated OM fraction is more labile than the non-Fe-associated fraction, as has been shown several

times in the past. If the more labile Fe-associated OM is removed from the total sample, then the residual, non-Fe-associated fraction will appear more refractory, as seen here (and as suggested on lines 129-131).

Authors' response:

We appreciate this comment.

Reviewer #1:

4. Lines 134: What "strong correlation"? Figure S2 shows the graphical results from a PCA analysis. Please explain.

Authors' response:

To clarify, we have now included linear regression analyses in the main material to demonstrate the strong correlation mentioned. Please check Fig 2.

Reviewer #1: 5. Lines 148-151: The OC:Fe ratio is not a reliable clue for the type of interactions between iron (oxyhydr)oxides and OC, as explained in Barber et al. (2017), because a fraction of reducible Fe is found unassociated with OC.

Authors' response:

We appreciate your comment and for raising this important discussion. Indeed, as highlighted by Barber et al. (2017), a portion of reducible Fe is not directly associated with OC. Consequently, we have revised our manuscript to remove the inference about the nature of this association. Nevertheless, the OC:Fe ratio remains a useful proxy for understanding the affinity of Fe phases with OC, and our findings are consistent with the existing literature. Specifically, data suggest that associations with poorly crystalline Fe oxides generally exhibit higher OC:Fe ratios compared to those with crystalline Fe oxides (Chen et al., 2014; Wagai & Mayer, 2007), which supports our results. We have updated the manuscript text to address this point more clearly (see lines 171-181).

Reviewer #1:

6. Lines 158-165: This is an elegant way of demonstrating what was already assumed (that OC associated predominantly with amorphous iron (oxyhydr)oxides).

Authors' response:

We appreciate the reviewer's comment.

Reviewer #1:

7. Lines 174-176: There seems to be a slight shift towards more oxidizing conditions indeed, but is this shift significant? Please provide the statistical support for this claim (or lack of support). Given the spread of the data, their overlap also, it is difficult to say whether this shift is statistically significant.

Authors' response:

We have performed PERMANOVA tests with the data. The results show that all areas differ regarding Eh, except Mangroves pristine and Mangroves adjacent to shrimp ponds. As to pH, only pasture differ from the mangroves.

Reviewer #1:

8. Line 187: What do these average and standard deviation correspond to? All datapoints for the pristine mangrove and mangrove adjacent to the shrimp ponds? Please explain.

Authors' response:

We appreciate the opportunity to clarify this point. We have indicated more clearly our results. Please see lines 203-213. We have also added the summary statistics of all measured variables as well as the PERMANOVA results in the "Extended data" file.

Reviewer #1:

9. Lines 187 and 188: Please provide an explanation and a standard deviation for the values for the shrimp ponds and the pasture.

Authors' response:

We thank the reviewer's comment. We have responded to this inquiry above. The revised text was added to lines 203-213.

Reviewer #1:

10. Lines 190-191: The first two values are not significantly different given the very large standard deviation for the shrimp ponds. Also, please provide a standard deviation for the pasture sample (again, it is a depth-average average, not a single datapoint).

Authors' response:

We thank the reviewer for the attention to this point. Indeed, the difference between the pristine Mangrove and shrimp pond was not significant (p -value = 0.42) due to the large variation in Shrimp pond. We included the standard deviation for pasture. Please note that the values reported are depth-averaged.

Reviewer #1:

11. Figure 4a: How do you explain the very high standard deviations found for the shrimp pond samples at each depth? Also, how do you explain the variations between different depths, also for the shrimp pond? You report a 6-fold decrease in Fe-citrate concentration between the 30-40 cm layer and the 40-50 cm layer – a huge decrease for two adjacent layers.

Authors' response:

We acknowledge the reviewer's inquiry. The high standard deviations observed in the shrimp pond samples at each depth are likely due to the drainage and refilling activities associated with shrimp pond management, as well as the death of roots due to deforestation and consequently impacting on soil aeration. These factors likely caused Fe segregation, leading to variability in Fe_{cbd} concentrations across different depths. This also explains the significant decrease in Fe_{cbd} concentration between the 30-40 cm and 40-50 cm layers, which is attributed to the heterogeneous distribution of Fe due to these disturbances. Despite these variations, our analysis highlights a very clear trend towards higher concentrations of Fe_{cbd} in shrimp ponds. We have updated the manuscript (lines 209-217) to better reflect these observations and clarify the depth-related variations.

Reviewer #1:

12. Figure 4b: Please add the standard deviations for the pasture sample since the values are depth-averaged.

Authors' response:

We have added the s.d. as suggested.

Reviewer #1:

13. Lines 207-208: Except for the pasture sample – please specify.

Authors' response:

We thank the attention to detail. We have revised the text as suggested.

Reviewer #1:

14. Line 219: Again, please add the standard deviation for the pasture sample here and in Figure 5c.

Authors' response:

We have added the s.d. as suggested. Please note that we have now reported the results as % of SOC, in place of % MAOC, to better depict the precise proportion of Fe-bound OC in the soils.

Reviewer #1:

15. Lines 259-260: This is entirely speculative as there is no data in this work that suggests oxidation of pyrite, aside from a decrease in pH that could have different causes. Please indicate that this is a speculative explanation in the text.

Authors' response:

We appreciate the reviewer's comment. To properly address this inquiry, we have performed further wet-chemical extractions on bulk samples that confirm our claim. The results are available in the Supplementary material (Fig S3). In fact, there was a sharp decrease in pyritic-Fe, as we hypothesized. We have mentioned the results in lines 263-264 and described the method in lines 444-457.

Reviewer #1:

16. Line 260: The soil pH is between 4 and 6, with an average of about 5, which is one order of magnitude higher in H⁺ concentration compared to what is written in the paper (which is misleading). On Figure 3, all but one or two datapoints fall into the Fe₂O₃ stability zone rather than the Fe²⁺ stability zone of the Pourbaix diagram. The pH is probably a little too high for this explanation to be the only reason for the Fe losses. Can you think of an alternative explanation?

Authors' response:

Thank you for your comment. The mean pH in the pasture soil was 4.5, which is 1.5–2 units lower than typical mangrove soils, as indicated in the data shared on Figshare (doi: 10.6084/m9.figshare.27111262). This lower pH is likely due to pyrite oxidation, a process known to significantly acidify soils.

Although the pH measured at the time of sampling was 4.5, it is possible that pH levels were lower during periods of active pyrite oxidation, resulting in more acidic conditions and potentially greater Fe losses. Acidic dissolution of iron oxides under these conditions could explain the Fe depletion observed, even though the pH was somewhat higher during the time of measurement.

Previous study have shown that a pH of 4.7 is sufficient to cause substantial Fe dissolution from iron oxides in environments subject to redox oscillations (Ferreira et al., 2022). Furthermore, while the Pourbaix diagram serves as a useful reference for understanding these processes, it is important to note that the boundaries of stability fields for Fe oxides are not sharp. Rather, transitions are gradual, meaning that processes such as Fe dissolution can still occur near the stability zones.

We have updated the text in lines 290-295.

Reviewer #1:

17. Line 288: This is a high sedimentation rate! Iron being a predominant element in soils of Amazonia (laterite soils), the particles carried by rivers must be enriched in this element, and therefore replenishing the soils' Fe content following remediation should not be a lengthy process. While the concluding remarks on lines 288-297 are true to some extent, they paint a darker picture than what the reality probably is. Maybe this should be mentioned?

Authors' response:

We appreciate the reviewer's observations and acknowledge the concerns raised about our interpretation of sedimentation rates and Fe replenishment in mangrove soils. We have revised the manuscript to reflect these considerations and provide a more nuanced discussion of the challenges and potential for Fe replenishment in the context of mangrove restoration.

However, it is important to emphasize that the Fe content in the studied mangrove soils, as well as in the sediments reaching these mangroves, is low. This is primarily due to the geological characteristics of the Barreiras Formation in which the surrounding upland soils are developed. These soils are typically highly weathered, with kaolinite as the dominant clay mineral and only minor amounts of Fe and Al oxides (Melo et

al., 2001). As a result, the mangrove soils in this region are also naturally low in Fe content (Andrade et al., 2018).

Please refer to the updated text in lines 309-328 for further details.

Reviewer #1:

18. Line 304: Please explain what "Am" means.

Authors' response:

The regional climate is tropical monsoon (Am), according to the Köppen classification. We have clarified this in the text (line 355).

Reviewer #1

19. Lines 333-334: This is not clear. The material retained on which one of the sieves? The first one only? Please be more explicit.

Authors' response:

We have clarified this point in the manuscript. All the particles retained in the sieves, i.e., particles > 20 µm) were collected and taken as POC, while the particles that passed through the 20 µm sieve were considered as MAOC. Please see lines 365-367.

Reviewer #1

20. Lines 337-338: Please provide the details for the centrifugation step.

Authors' response:

We appreciate the opportunity to clarify this point. The suspensions were centrifuged at 6,000 x g for 10 minutes. We have added this information in line 370.

Reviewer #1

21. Line 339: Freeze-drying changes the crystallinity of iron (oxyhydr)oxides, and thus its extractability with the different complexation or reducing agents used in this work. Have you run any before/after freeze-drying controls? How do you know then whether the data obtained after freeze-drying the samples is representative of the in situ samples? Please address this point carefully as it could have led to significant changes in the distribution of OC among the different Fe fractions.

Authors' response:

We thank the reviewer again for raising this concern. We have addressed this inquiry in detail in a previous response. Please, see lines 426-435 and Tables S3 and S4.

Reviewer #1:

22. Lines 352-373: Please provide the volumes used for the extractions. Also, how did you ascertain that the complexation capacity of Na-citrate at 0.2 M was not surpassed since the initial mass of sediment sieved was 20 g ? The same is true for the other extractions as well (complexation capacity and reduction capacity for ascorbate and dithionite).

Authors' response:

We appreciate the reviewer's suggestion and have added further details to the procedure. The extraction process involved sequential treatments with Na-citrate (Fe_{cit}), citrate-ascorbate (Fe_{asc}), and citrate-bicarbonate dithionite (Fe_{cbd}). Specifically, 40 ml of each solution was added to 0.50 g of freeze-dried, ground samples in 50 ml centrifuge tubes. The volume:sample ratio of 80:1 is deemed sufficient for the complete extraction of Fe phases (Lalonde et al., 2012). We have included these details in lines 388-391.

Reviewer #1:

23. Lines 402-411: Please explain how the samples were prepared for FTIR analysis since the absolute peak intensities are compared between samples. Surely there must have been some sort of normalization? Lower intensities could simply mean less abundant organic matter in the OM-KBr mix rather than a lower relative abundance of a given functional group.

Authors' response:

We appreciate the opportunity to give more details regarding the FTIR treatments. The sample was analyzed as KBr pellets and spectra were obtained from 4000 to 400 cm^{-1} at a resolution of 1 cm^{-1} . Each sample was scanned 64 times and averaged to generate the final spectrum. The spectra were baseline corrected, normalized (min-max) and peak area of the regions of interest were calculated without smoothing. The spectra were subsequently smoothed for improved visual presentation using the Savitzky-Golay method with a quadratic function and 11 data points. All these procedures were done using the Orange 3.36 software. This description is found in lines 461-468.

Reviewer #1

24. Table S1, Fe concentration column: The data should have the same number of decimal points throughout. The number in the second decimal point should be placed in subscript to indicate that it is indicative only rather than significant.

Authors' response:

We thank the reviewer's attention to this. We have added the second decimal number to the Fe results.

Reviewer #1

25. Figure S2: More explanation about what we are seeing on this PCA graph should be given in the figure caption.

Authors' response:

We appreciate this suggestion. We chose to show the linear regressions between the variables of interest instead of using the PCA. We believe that this more clearly illustrates our point. Please refer to Fig 2 in the main file.

We sincerely thank the reviewer for this positive feedback on our manuscript. We are pleased to hear that the paper was found to be well-written and accessible to a broader audience. We have carefully addressed all the points raised and appreciate the time and effort taken to review our work. We hope the revisions now meet the reviewer's expectations.

Reviewer #2: The research presented by Ruiz et al. investigates the role of Fe in SOC (De)stabilization in mangrove forests under LUC. The research is of environmental relevance, however does not add significantly to the existing literature. It is a good study, but in my opinion does not meet the novelty criteria to be published in Nature Communications. I wish you all the best to the team and look forward to reading more about this research in future.

Authors' response:

We appreciate the reviewer's feedback and the recognition of our research's environmental relevance. While we value the reviewer's perspective, we respectfully disagree with the assessment regarding the novelty of our work. We would like to highlight key aspects that we believe significantly advance the current understanding of SOC dynamics in mangrove ecosystems.

Original Contribution to Fe mediated Interactions (FeOMs) in Mangrove Soils: While the role of Fe in stabilizing SOC has been explored in terrestrial ecosystems, our study is the first to examine these dynamics specifically in mangrove soils. To the best of our knowledge only two previous studies assessed Fe-bound OC but it was restricted to quantify this pool (Assavapanuvat et al., 2024; Dicen et al., 2019), and did not assess its nature and significance for SOC stabilization. Moreover, the biogeochemical interactions between Fe and SOC in Amazonian mangroves, especially under LUC scenarios like conversion to shrimp ponds or pastures, remain poorly understood. Our findings on FeOMs in these environments address this significant knowledge gap, suggesting new pathways for SOC (de)stabilization that have not been previously reported.

Novelty in methodological approach: Our work employs a combination of wet chemical extractions, thermal (TG-DSC) analysis, and spectroscopy (FTIR) analysis to investigate Fe-SOC interactions. This approach is not restricted to mangrove soils and can be replicated to upland soils, offering a deeper understanding of the different organomineral interactions that govern SOC persistence across diverse ecosystems.

Environmental implications: The rapid rate of mangrove loss and conversion to other land uses is a global environmental concern. By addressing how these changes impact SOC dynamics, particularly through the lens of Fe interactions, our study provides important data that can inform conservation and restoration strategies not only in the Amazon but also in other mangrove regions worldwide. This has broader implications for global carbon budgets and climate change mitigation efforts.

To further illustrate the novelty of our work, we conducted a search on the Scopus platform. When including all soil types in research related to organomineral interactions with Fe, 213 articles were found. However, when "mangrove" was added as a keyword, the results dropped to just one article. This demonstrates the uniqueness of our study, which not only focuses on mangrove soils but also delves into the mechanisms and stability of Fe-C associations, an area that previous studies have not explored in the same depth.

Analyze search results

Analyze search results

We believe these contributions justify the novelty of our work and its suitability for publication in Nature Communications. We thank the reviewer once again for his/her valuable feedback and encouragement.

Reviewer #3:

My first concern is that the massive loss of SOC due to LUC was not supported by the mechanism of FeOMI, as proposed by the authors. Conversion of mangroves to shrimp pond and pasture resulted in substantial loss of SOC, especially for the MAOC fractions. Although loss of Fe-bound OC might be important in mangrove ecosystem, it's not sufficient to explain the contribution of FeOMI in loss of MAOC. Providing direct evidence that crystalline iron retains less soil organic carbon would strongly support the authors' viewpoint

Authors' response:

We appreciate the reviewer's insightful comment and the opportunity to clarify our findings.

We acknowledge that the loss of Fe-bound OC is not the sole reason for the substantial loss MAOC following land-use change. We focus on Fe-bound OC and we show that the conversion and loss of Fe phases, especially those related to poorly crystalline Fe oxides is a major reason for the loss of OC. Importantly we demonstrate that FeOMI associations, which are generally considered stable, are actually highly susceptible to changes in the geochemical environment resulting from LUC.

We demonstrate the importance of poorly crystalline Fe oxides by presenting the OC:Fe ratio from our sequential extraction results (Table S2). Specifically, the Fe extracted by citrate-ascorbate (Fe asc) showed at least twice as much OC relative to Fe compared to the Fe extracted by citrate-bicarbonate-dithionite (Fe cbd). This finding underscores our point about the greater efficiency of poorly crystalline Fe oxides in binding OC. Our extraction data also revealed that the OC:Fe ratio for Fe asc was three times greater than for Fe cbd,

implying that both adsorption and coprecipitation mechanisms are involved in FeOMs with poorly crystalline Fe oxides.

To further address the reviewer's request for direct evidence, we conducted an additional adsorption and coprecipitation experiment with goethite and ferrihydrite reacting with humic acid (HA). The detailed methods and results are available in the Supplementary material. The results showed that ferrihydrite adsorbed approximately 65 mg HA g⁻¹, while goethite adsorbed only 27 mg HA g⁻¹, indicating that ferrihydrite binds three times more OC than goethite. Moreover, in the coprecipitation experiments, the mean HA coprecipitated with ferrihydrite was approximately 690 mg HA g⁻¹, ten times greater than what was observed in the adsorption experiment.

We hope that these new data provide the additional evidence required to support our viewpoint and address the reviewer's concerns.

We have updated lines 168-181 to include this discussion.

Reviewer #3:

"I agree with first result regarding the critical role of FeOMI in mangrove soils and the explanation that iron binds more labile carbon. However, the evidence for the conclusion that Fe oxides protect more labile SOC fractions was not sufficient. It is plausible that Fe oxides affinity for more thermally labile and oxidized organic compounds. But it can also not exclude the explanation that catalytic effect of Fe oxides on organic matter oxidation during thermal analysis. In addition, thermally labile OC is not equivalent to labile SOC fraction. It's better to provide direct evidence that how Fe oxides protect POC and MAOC."

Authors' response:

Thank you for this important inquiry. We appreciate your recognition of the critical role of Fe-organomineral interactions (FeOMI) in mangrove soils and we acknowledge the need for further clarification regarding our conclusion that Fe oxides protect more labile SOC fractions.

The reviewer raises an important question about the potential catalytic effect of Fe oxides on OM oxidation during thermal analysis. However, when we remove Fe and the associated OM, the remaining OM is still associated with other minerals, which can also produce a catalytic effect (Dembicki, 1992; Faure et al., 2006). Therefore, it is less likely that the observed effect is primarily due to catalysis and more likely due to OM composition in these different MAOC subfractions, where Fe-bound OC is more thermally labile.

This interpretation is supported by the "bioenergetic framework" (Williams et al., 2018; Williams & Plante, 2018) and the current paradigm of SOM persistence (Kleber et al., 2011; Lehmann et al., 2020; Lehmann & Kleber, 2015). This perspective suggests that SOM exists as a continuum, continuously processed by the decomposer community from large biopolymers to small monomers. As these compounds become increasingly oxidized and soluble, they become more susceptible to microbial uptake unless they are protected from decomposition through mineral protection (Kleber et al., 2015).

Depolymerization and oxidation of larger organic molecules solubilize smaller organic molecules, which make them more accessible for microbial uptake if the dissolved organic matter is readily available (Lehmann et al., 2020). On the other hand, oxidation enhances chemical reactivity of organic compounds by reducing molecular size, increasing polarity, and thus increasing chemical reactivity. This is crucial for the

formation of organomineral interactions, especially those involving Fe, which include ligand exchange with oxygen-bearing functional groups (Chen et al., 2014; Gu et al., 1995).

When compared to POC, MAOC has lower thermal stability, reflected by lower energy density (ED) and lower activation energy, the latter positively correlated with DSC-T₅₀ (Williams et al., 2018; Williams & Plante, 2018). This is because MAOC is composed of easily metabolizable organic molecules with low thermodynamic stability (e.g., high O-alkyl C/aromatic C ratio, a higher proportion of thermally labile materials). Thus, a decrease in thermal stability is expected as SOM decomposition progresses, with larger plant biopolymers being processed into smaller molecules (Barré et al., 2016).

Although MAOC generally exhibits relatively low thermal stability, different stability is observed within this fraction when different types of bonds occur. For instance, increasing cation-bridging associations lead to higher thermal resistance of adsorbed organic matter (Barreto et al., 2021). Ligand exchange, one of the primary mechanisms of OM bonding with Fe oxides, can have relatively large activation energy for adsorption (Nguyen et al., 2018). Based on this rationale, we would expect the DSC-T₅₀ to decrease after Fe removal, as strong covalent ligand-exchange reactions would become less frequent than the weaker bonding (e.g., electrostatic, van der Waals) more common with the remaining phyllosilicates (Kleber et al., 2021). However, we observed the opposite. Thus, a more plausible explanation for the relative increase in thermal stability after the removal of Fe-bound OC is that this organic fraction had lower thermal stability than the remaining one.

Our FTIR data further supports this interpretation. Upon removing Fe-associated OC, the COO/CH ratio decreases, indicating that the more oxidized organic matter (i.e., more decomposed and likely biologically labile) is removed along with Fe. In fact, there is a positive correlation between DSC-T₅₀ and the CH/COO⁻ ratio, meaning that as the CH/COO⁻ ratio increases (i.e., Fe-bound OC is removed), the remaining MAOC becomes more thermally resistant (see Fig r1 below), which can be attributed to the increased relative content of energetic C—H bonds (Williams et al., 2018). This implies that within the MAOC fraction, there is a subfraction (i.e., Fe-associated) that is thermally and inherently biologically more labile.

Another important piece of evidence we included in our manuscript is the increase in the C/N ratio as Fe-bound OC is removed, suggesting that the organic matter remaining in the MAOC fraction is less decomposed and more chemically complex than that associated with Fe. In fact, we observed a positive correlation between the C/N ratio and ED ($R^2_{\text{adj}} = 0.39$; p-value = 0.02). This further supports our argument that the effect of organic matter composition on thermal stability is more important than any catalysis reaction due to Fe mineral. Therefore, we can conclude that Fe-bound OC is more decomposed and less complex than the average MAOC, indicating that Fe helps protect an inherently more biologically labile SOC fraction.

We have edited lines 118-149 to better clarify the evidence and rationale of our claim.

We hope this explanation addresses your concerns and clarifies our conclusions regarding the role of Fe oxides in protecting *a priori* labile SOC fractions in mangrove soils.

Reviewer #3:

In the introduction, it is necessary to briefly describe the important role of mineral protection mechanisms in the stabilization of soil organic carbon in mangroves. The relationship between the forms of iron, or their crystallinity, and soil organic carbon should be clearly introduced. Additionally, hypotheses about the impact of land use change on iron and the iron-OM bound relationship should be reviewed with published literature.

Authors' response:

We appreciate the reviewer's feedback on the Introduction section. We have revised it to better reflect the current understanding of FeOMs and to clarify our hypotheses. Please see the revised text in lines 60-77.

Reviewer #3:

Table S1, which is a key result, is somewhat difficult to understand. The parameters displayed in the table need clarification. It appears that some indicators refer to the extracted portion while others refer to the residual portion. This should be more clearly indicated in the table explanation.

Authors' response:

Thank you for your valuable feedback. We have revised the table legend to clearly distinguish between the parameters related to the extracted portion and those related to the residual portion.

Reviewer #3:

The discussion on the environmental implications does not seem to independent part. This part does not present any new findings or conclusions. I do not understand what the authors intend to express about the role of the environment in carbon sequestration in mangroves and the iron-OC binding relationship under LUC. Additionally, it is unclear what is meant by "environment". Supplementing this section with specific facts and relevant data would make it more compelling.

Authors' response:

We appreciate the reviewer's comment. To address the concern that the environmental implications section lacks independence and clarity, we have revised the text to more explicitly discuss the role of environmental factors—specifically, sediment deposition and Fe dynamics—in the carbon sequestration process in mangrove ecosystems under land-use change (LUC). We have renamed the section title to "Implications for future restoration efforts" to better reflect its intended focus.

Reviewer #3:

There also some other minor errors in the text and figures, such as the lack of a b annotation in Fig.2, and the Table 1 mentioned in L133 is Table S1. I recommend that the authors thoroughly review the entire text to avoid these errors which easily make reader confusing.

Authors' response:

Thank you for pointing out these issues. We have revised the entirely text, including Table and Fig references.

We would like to thank you for your detailed and helpful feedback. We have revised the manuscript accordingly and are confident that we have addressed all the issues raised by the reviewer. Thank you for your time and effort, which have greatly contributed to enhancing the quality of our work.

References

- Andrade, G. R. P., Cuadros, J., Partiti, C. S. M., Cohen, R., & Vidal-Torrado, P. (2018). Sequential mineral transformation from kaolinite to Fe-illite in two Brazilian mangrove soils. *Geoderma*, 309, 84–99. <https://doi.org/https://doi.org/10.1016/j.geoderma.2017.08.042>
- Araújo Júnior, J. M. de C., Ferreira, T. O., Suarez-Abelenda, M., Nóbrega, G. N., Albuquerque, A. G. B. M., Bezerra, A. de C., & Otero, X. L. (2016). The role of bioturbation by *Ucides cordatus* crab in the fractionation and bioavailability of trace metals in tropical semiarid mangroves. *Marine Pollution Bulletin*, 111(1–2), 194–202. <https://doi.org/10.1016/j.marpolbul.2016.07.011>

- Assavapanuvat, P., Breithaupt, J. L., Engelbert, K. M., Schröder, C., Smoak, J. M., & Bianchi, T. S. (2024). Contrasting stocks and origins of particulate and mineral-associated soil organic carbon in a mangrove-salt marsh ecotone. *Geoderma*, *446*. <https://doi.org/10.1016/j.geoderma.2024.116904>
- Barré, P., Plante, A. F., Cécillon, L., Lutfalla, S., Baudin, F., Bernard, S., Christensen, B. T., Eglin, T., Fernandez, J. M., & Houot, S. (2016). The energetic and chemical signatures of persistent soil organic matter. *Biogeochemistry*, *130*, 1–12.
- Barreto, M. S. C., Elzinga, E. J., Ramlogan, M., Rouff, A. A., & Alleoni, L. R. F. (2021). Calcium enhances adsorption and thermal stability of organic compounds on soil minerals. *Chemical Geology*, *559*, 119804. <https://doi.org/https://doi.org/10.1016/j.chemgeo.2020.119804>
- Boone Kauffman, J., Arifanti, V. B., Hernández Trejo, H., del Carmen Jesús García, M., Norfolk, J., Cifuentes, M., Hadriyanto, D., & Murdiyarso, D. (2017). The jumbo carbon footprint of a shrimp: carbon losses from mangrove deforestation. *Frontiers in Ecology and the Environment*, *15*(4), 183–188. <https://doi.org/https://doi.org/10.1002/fee.1482>
- Chen, C., Dynes, J. J., Wang, J., & Sparks, D. L. (2014). Properties of Fe-Organic Matter Associations via Coprecipitation versus Adsorption. *Environmental Science & Technology*, *48*(23), 13751–13759. <https://doi.org/10.1021/es503669u>
- Dembicki, H. (1992). The effects of the mineral matrix on the determination of kinetic parameters using modified Rock Eval pyrolysis. *Organic Geochemistry*, *18*(4), 531–539. [https://doi.org/https://doi.org/10.1016/0146-6380\(92\)90116-F](https://doi.org/https://doi.org/10.1016/0146-6380(92)90116-F)
- Dicen, G. P., Navarrete, I. A., Rallos, R. V., Salmo, S. G., & Garcia, M. C. A. (2019). The role of reactive iron in long-term carbon sequestration in mangrove sediments. *Journal of Soils and Sediments*, *19*(1), 501–510. <https://doi.org/10.1007/s11368-018-2051-y>
- Faure, P., Schlepp, L., Mansuy-Huault, L., Elie, M., Jardé, E., & Pelletier, M. (2006). Aromatization of organic matter induced by the presence of clays during flash pyrolysis-gas chromatography–mass spectrometry (PyGC–MS): A major analytical artifact. *Journal of Analytical and Applied Pyrolysis*, *75*(1), 1–10. <https://doi.org/https://doi.org/10.1016/j.jaap.2005.02.004>
- Ferreira, A. D., Queiroz, H. M., Otero, X. L., Barcellos, D., Bernardino, Â. F., & Ferreira, T. O. (2022). Iron hazard in an impacted estuary: Contrasting controls of plants and implications to phytoremediation. *Journal of Hazardous Materials*, *428*, 128216. <https://doi.org/https://doi.org/10.1016/j.jhazmat.2022.128216>
- Gu, B., Schmitt, J., Chen, Z., Liang, L., & McCarthy, J. F. (1995). Adsorption and desorption of different organic matter fractions on iron oxide. *Geochimica et Cosmochimica Acta*, *59*(2), 219–229. [https://doi.org/https://doi.org/10.1016/0016-7037\(94\)00282-Q](https://doi.org/https://doi.org/10.1016/0016-7037(94)00282-Q)
- Kauffman, J. B., Hernandez Trejo, H., del Carmen Jesus Garcia, M., Heider, C., & Contreras, W. M. (2016). Carbon stocks of mangroves and losses arising from their conversion to cattle pastures in the Pantanos de Centla, Mexico. *Wetlands Ecology and Management*, *24*(2), 203–216. <https://doi.org/10.1007/s11273-015-9453-z>
- Kleber, M., Bourg, I. C., Coward, E. K., Hansel, C. M., Myneni, S. C. B., & Nunan, N. (2021). Dynamic interactions at the mineral–organic matter interface. In *Nature Reviews Earth and Environment* (Vol. 2, Issue 6, pp. 402–421). Springer Nature. <https://doi.org/10.1038/s43017-021-00162-y>

- Kleber, M., Eusterhues, K., Keilluweit, M., Mikutta, C., Mikutta, R., & Nico, P. S. (2015). *Chapter One - Mineral–Organic Associations: Formation, Properties, and Relevance in Soil Environments* (D. L. B. T.-A. in A. Sparks, Ed.; Vol. 130, pp. 1–140). Academic Press. <https://doi.org/https://doi.org/10.1016/bs.agron.2014.10.005>
- Kleber, M., Nico, P. S., Plante, A., Filley, T., Kramer, M., Swanston, C., & Sollins, P. (2011). Old and stable soil organic matter is not necessarily chemically recalcitrant: implications for modeling concepts and temperature sensitivity. *Global Change Biology*, *17*(2), 1097–1107.
- Lalonde, K., Mucci, A., Ouellet, A., & Gélinas, Y. (2012). Preservation of organic matter in sediments promoted by iron. *Nature*, *483*(7388), 198–200. <https://doi.org/10.1038/nature10855>
- Lehmann, J., Hansel, C. M., Kaiser, C., Kleber, M., Maher, K., Manzoni, S., Nunan, N., Reichstein, M., Schimel, J. P., Torn, M. S., Wieder, W. R., & Kögel-Knabner, I. (2020). Persistence of soil organic carbon caused by functional complexity. *Nature Geoscience*, *13*(8), 529–534. <https://doi.org/10.1038/s41561-020-0612-3>
- Lehmann, J., & Kleber, M. (2015). The contentious nature of soil organic matter. *Nature*, *528*(7580), 60–68. <https://doi.org/10.1038/nature16069>
- Melo, V. F., Singh, B., Schaefer, C. E. G. R., Novais, R. F., & Fontes, M. P. F. (2001). Chemical and Mineralogical Properties of Kaolinite-Rich Brazilian Soils. *Soil Science Society of America Journal*, *65*(4), 1324–1333. <https://doi.org/https://doi.org/10.2136/sssaj2001.6541324x>
- Nguyen, M. L., Hockaday, W. C., & Lau, B. L. T. (2018). Is the adsorption of soil organic matter to haematite (α -Fe₂O₃) temperature dependent? *European Journal of Soil Science*, *69*(5), 892–901. <https://doi.org/https://doi.org/10.1111/ejss.12694>
- Otero, X. L., Ferreira, T. O., Vidal-Torrado, P., & Macías, F. (2006). Spatial variation in pore water geochemistry in a mangrove system (Pai Matos island, Cananea-Brazil). *Applied Geochemistry*, *21*(12), 2171–2186. <https://doi.org/https://doi.org/10.1016/j.apgeochem.2006.07.012>
- Wagai, R., & Mayer, L. M. (2007). Sorptive stabilization of organic matter in soils by hydrous iron oxides. *Geochimica et Cosmochimica Acta*, *71*(1), 25–35. <https://doi.org/10.1016/j.GCA.2006.08.047>
- Williams, E. K., Fogel, M. L., Berhe, A. A., & Plante, A. F. (2018). Distinct bioenergetic signatures in particulate versus mineral-associated soil organic matter. *Geoderma*, *330*, 107–116. <https://doi.org/https://doi.org/10.1016/j.geoderma.2018.05.024>
- Williams, E. K., & Plante, A. F. (2018). A Bioenergetic Framework for Assessing Soil Organic Matter Persistence. *Frontiers in Earth Science*, *6*. <https://www.frontiersin.org/articles/10.3389/feart.2018.00143>